# CDC20 promotes bone formation via APC/C dependent ubiquitination and degradation of p65

Yangge Du [iD], Min Zhang [iD], Xuejiao Liu [iD], Zheng Li [iD], Menglong Hu [iD], Yueming Tian [iD], Longwei Lv [iD], Xiao Zhang [iD], Yunsong Liu [iD], Ping Zhang* & Yongsheng Zhou** [iD]

## Abstract

The E3 ubiquitin ligase complex CDC20-activated anaphase-promoting complex/Cyclosome (APC/C$^{CDC20}$) plays a critical role in governing mitotic progression by targeting key cell cycle regulators for degradation. Cell division cycle protein 20 homolog (CDC20), the co-activator of APC/C, is required for full ubiquitin ligase activity. In addition to its well-known cell cycle-related functions, we demonstrate that CDC20 plays an essential role in osteogenic commitment of bone marrow mesenchymal stromal/stem cells (BMSCs). Cdc20 conditional knockout mice exhibit decreased bone formation and impaired bone regeneration after injury. Mechanistically, we discovered a functional interaction between the WD40 domain of CDC20 and the DNA-binding domain of p65. Moreover, CDC20 promotes the ubiquitination and degradation of p65 in an APC11-dependent manner. More importantly, knockdown of p65 rescues the bone loss in Cdc20 conditional knockout mice. Our current work reveals a cell cycle-independent function of CDC20, establishes APC11$^{CDC20}$ as a pivotal regulator for bone formation by governing the ubiquitination and degradation of p65, and may pave the way for treatment of bone-related diseases.

**Keywords** APC/C; BMSCs; CDC20; osteogenic differentiation; p65
**Subject Categories** Development; Musculoskeletal System; Post-translational Modifications & Proteolysis

## Introduction

APC/C is a multifunctional ubiquitin-protein ligase targeting various substrates for proteolysis outside and inside the cell cycle (Qiao *et al*, 2010). APC/C also has cell cycle-independent functions, which includes regulation of genomic integrity and cell differentiation of the nervous system (Wäsch *et al*, 2010; Puram & Bonni, 2011). APC/C is composed of 14–16 different subunits and exhibits a unique molecular architecture. The structure of APC/C comprises platform, TPR lobe, and catalytic core. The catalytic core consists of the RING domain subunit APC11 and the Cullin domain subunit APC2 (Gmachl *et al*, 2000). The flexibility of the catalytic module is important for APC/C functions and is influenced by the orientation of the platform and by direct interaction partners of APC11/APC2. APC2 and APC11 formed a complex in the absence of other APC/C subunits, comprising the minimal ubiquitin ligase module of the APC/C (Tang *et al*, 2001). The APC/C core associates with the substrate-recruiting proteins CDC20 and CDH1 to form two functionally E3 ligase complexes, APC/C$^{CDC20}$ and APC/C$^{CDH1}$, respectively. The co-activator subunit (CDC20 or CDH1) plays a significant role in specifying substrate recognition and enhancing the catalytic activity of APC/C (Kimata *et al*, 2008; Van Voorhis & Morgan, 2014). CDC20 and CDH1 are characterized by having the WD40 domain structure, mediating direct binding to the APC/C (Izawa & Pines, 2012; Charchaflieh *et al*, 2015). CDH1 plays a crucial role in controlling cell cycle progression, genomic integrity, DNA damage repair, and cellular metabolism (Peters, 2006; Gao *et al*, 2009; Herrero-Mendez *et al*, 2009). In mitosis, CDH1 functions mainly during mitotic exit, while CDC20 plays a key role in early mitosis of cell cycle and leads to the ubiquitination and degradation of specific substrates of APC/C. Many studies have reported the biological functions of CDC20 in brain development (Yang *et al*, 2009), ciliary disassembly (Wang *et al*, 2014), and regulating cellular apoptosis (Wan *et al*, 2014). The expression of CDC20 has been reported to be elevated in higher grades of cancers and has been linked to poor prognosis in lung, bladder, colon, breast cancer, and oral squamous cell carcinomas (Chang *et al*, 2012; Choi *et al*, 2013; Moura *et al*, 2014). However, the effect of CDC20 on bone formation is not clear.

Characterized by their multi-potential differentiation capabilities and immune-regulatory properties, BMSCs have generated great interest in cell-based therapies for various congenital and acquired bone defects (Friedenstein *et al*, 1968). Bone remodeling is a process in which the osteoblast-mediated bone formation is closely coupled with osteoclast-mediated bone resorption. The E3 ligases have been reported to play important roles in the dynamic bone formation and resorption processes. The E3 ubiquitin ligase CHIP interacts with Runx2 and increases Runx2 degradation in preosteoblasts, leading to inhibition of osteoblast differentiation (Li *et al*,

Department of Prosthodontics, Peking University School and Hospital of Stomatology, National Engineering Laboratory for Digital and Material Technology of Stomatology, National Clinical Research Center for Oral Diseases, Beijing Key Laboratory of Digital Stomatology, Beijing, China
*Corresponding author. Tel: +86 10 8219 5370; E-mail: zhangping332@hsc.pku.edu.cn
**Corresponding author. Tel: +86 10 8219 5370; E-mail: kqzhouysh@hsc.pku.edu.cn

2008). Another study has claimed that the extinction of the E3 ubiquitin ligase c-Cbl expression decreases osteoclast adhesion, motility, and resorbing activity through modulating Src kinase degradation (Sanjay *et al*, 2001; Bruzzaniti *et al*, 2005). The E3 ubiquitin ligase Schnurri-3 deficiency in mice promotes osteoblast activity, and the E3 ubiquitin ligase Wwp1 deletion in mice results in increased Runx2 expression and bone mass (Jones *et al*, 2006). Most importantly, it has been demonstrated that one APC/C activator, CDH1 but not CDC20, induces osteoblast differentiation through governing Smurf1 E3 ligase activity and the downstream MEKK pathway (Wan *et al*, 2011). Then, we want to know whether CDC20 also plays a role in osteogenic differentiation. In the present study, we detected the pivotal role of CDC20 in the fate commitment of BMSCs for the first time. Conditional knockout of Cdc20 in mice leads to reduced bone mass and impaired bone regeneration after injury. Mechanistically, we demonstrated that APC11$^{CDC20}$ promoted the osteogenic differentiation of BMSCs by ubiquitination and degradation of p65. This research provides a new insight into how the APC11$^{CDC20}$ manipulates osteogenic differentiation of BMSCs, which will effectively exploit the potentiality of APC11$^{CDC20}$ in bone regeneration.

## Results

### Conditional knockout of Cdc20 impairs bone formation and blunts bone regeneration

To investigate the role of CDC20 on bone formation *in vivo*, we constructed the conditional knockout mice model. First, we generated Cdc20 floxed (*Cdc20$^{f/+}$*) mice using CRISPER/Cas9. Cre recombinase-mediated removal of exon is expected to lead a translation termination of Cdc20. Then, we bred *Cdc20$^{f/+}$* mice with *Sp7-Cre* transgenic mice to conditionally delete Cdc20 from skeletal tissues (Fig EV1A). The genotypes of transgenic mice were determined by DNA electrophoresis, and *Cdc20$^{f/f}$* were control groups, while *Sp7-Cre; Cdc20$^{f/f}$* were experimental groups (Fig EV1B). Micro-CT analyses and hematoxylin and eosin (H&E) staining of the distal femur metaphysis of *Sp7-Cre; Cdc20$^{f/f}$* mice showed an impairment in trabecular bone micro-architecture. The BMD and BV/TV parameters in *Sp7-Cre; Cdc20$^{f/f}$* 6-week-old male and female mice were greatly lower than their *Cdc20$^{f/f}$* littermates. *Sp7-Cre; Cdc20$^{f/f}$* mice also had decreased Tb.N and increased Tb. Sp (Fig 1A–D). Similar results in 12-week-old male and female mice were presented in Fig EV1C–F separately. To better characterize the skeletal

phenotype of conditional knockout mice, additional controls were presented. We investigated the skeletal phenotypes of *Sp7-Cre, Cdc20$^{f/f}$*, *Sp7-Cre; Cdc20$^{f/+}$*, and *Sp7-Cre; Cdc20$^{f/f}$* 6-week-old mice. Micro-CT analyses and histomorphometric measurements of the distal femur metaphysis of *Sp7-Cre; Cdc20$^{f/f}$* mice showed an impairment in trabecular bone compared to other controls. There were no significant differences among *Sp7-Cre, Cdc20$^{f/f}$*, *Sp7-Cre; Cdc20$^{f/+}$* phenotypes (Appendix Fig S1A–E). The identification of *Sp7-Cre, Cdc20$^{f/f}$*, *Sp7-Cre; Cdc20$^{f/+}$*, and *Sp7-Cre; Cdc20$^{f/f}$* genotypes was determined by DNA electrophoresis (Appendix Fig S1F). In addition to expression in osteoprogenitors, Sp7-Cre is also expressed in prehypertrophic and hypertrophic chondrocytes (Rodda & McMahon, 2006). Therefore, we conducted Goldner's trichrome staining to examine the growth plate phenotypes of 6-week-old mice and evaluated the lengths of growth plates through ImageJ software. The results showed that there were no significant differences between the lengths of growth plates of *Cdc20$^{f/f}$* and *Sp7-Cre; Cdc20$^{f/f}$* mice (Appendix Fig S2A and B). To figure out the endochondral bone formation as well as cartilage formation, we collected the femurs and tibiae of embryonic day 19 (E19) and postnatal day 1 (P1), day 4 (P4) mice and stained with H&E and Safranin-O-Fast Green. There were no discernable differences among the endochondral bone development in embryos and newborn mice (Appendix Fig S2C and D), indicating that the phenotype of bone loss was not due to altered cartilage formation. To further gain insight into the whole phenotypes of newborn mice skeletons, we performed Alcian Blue and Alizarin red staining of *Cdc20$^{f/f}$* and *Sp7-Cre; Cdc20$^{f/f}$* postnatal day 2 (P2) mice. The whole skeleton, upper extremities, and hind limbs were gathered. No discernable differences were seen in the staining of skeleton (Appendix Fig S2E). Then, we attempted to figure out the cellular causes of the osteopenia and the 6-week-old mice femurs were examined. We used markers osteocalcin (OCN) together with collagen type I (COL1) in immunohistochemistry (IHC) assay to characterize osteoblasts and tartrate-resistant acid phosphatase (TRAP) staining to characterize osteoclasts. The measurements of osteoblasts and osteoclasts were presented as N. Ob/B. Pm (osteoblast number/bone perimeter) or N. Oc/B. Pm (osteoclast number/bone perimeter), and no significant differences were seen in *Cdc20$^{f/f}$* and *Sp7-Cre; Cdc20$^{f/f}$* mice (Appendix Fig S3A and B). Then, we tried to clarify the proliferation ability of *Cdc20$^{f/f}$* and *Sp7-Cre; Cdc20$^{f/f}$* mice. According to 5-Ethynyl-2′-deoxyuridine (EdU) assay, the proliferating cells of control and conditional knockout mice showed no significant differences (Appendix Fig S3C). Additionally, the serum levels of

**Figure 1. Conditional knockout of Cdc20 impairs bone formation and blunts bone regeneration.**

A   Representative micro-CT images and H&E staining of trabecular bone from the femoral metaphysis of 6-week-old male *Sp7-Cre;Cdc20$^{f/f}$* and littermate control mice. Scale bar, 500 μm.

B   Histomorphometric analyses of 6-week-old male femurs (*n* = 6).

C   Representative micro-CT images and H&E staining of trabecular bone from the femoral metaphysis of 6-week-old female *Sp7-Cre;Cdc20$^{f/f}$* and littermate control mice. Scale bar, 500 μm.

D   Histomorphometric analyses of 6-week-old female femurs (*n* = 6).

E   Representative micro-CT images of tibial cortical bone defects in *Sp7-Cre;Cdc20$^{f/f}$* and littermate control mice. The green lines show the positions of the original defects. The green circles represent the regenerated bone. Scale bar, 500 μm.

F   Histomorphometric analysis of the regenerated bone in tibial cortical gaps (*n* = 5).

Data information: Data are displayed as mean ± SD and show one representative of *n* ≥ 3 independent experiments with three biological replicates. Statistical significance was calculated by a two-tailed unpaired Student's *t*-test defined as \*P < 0.05; \*\*P < 0.01; \*\*\*P < 0.001.

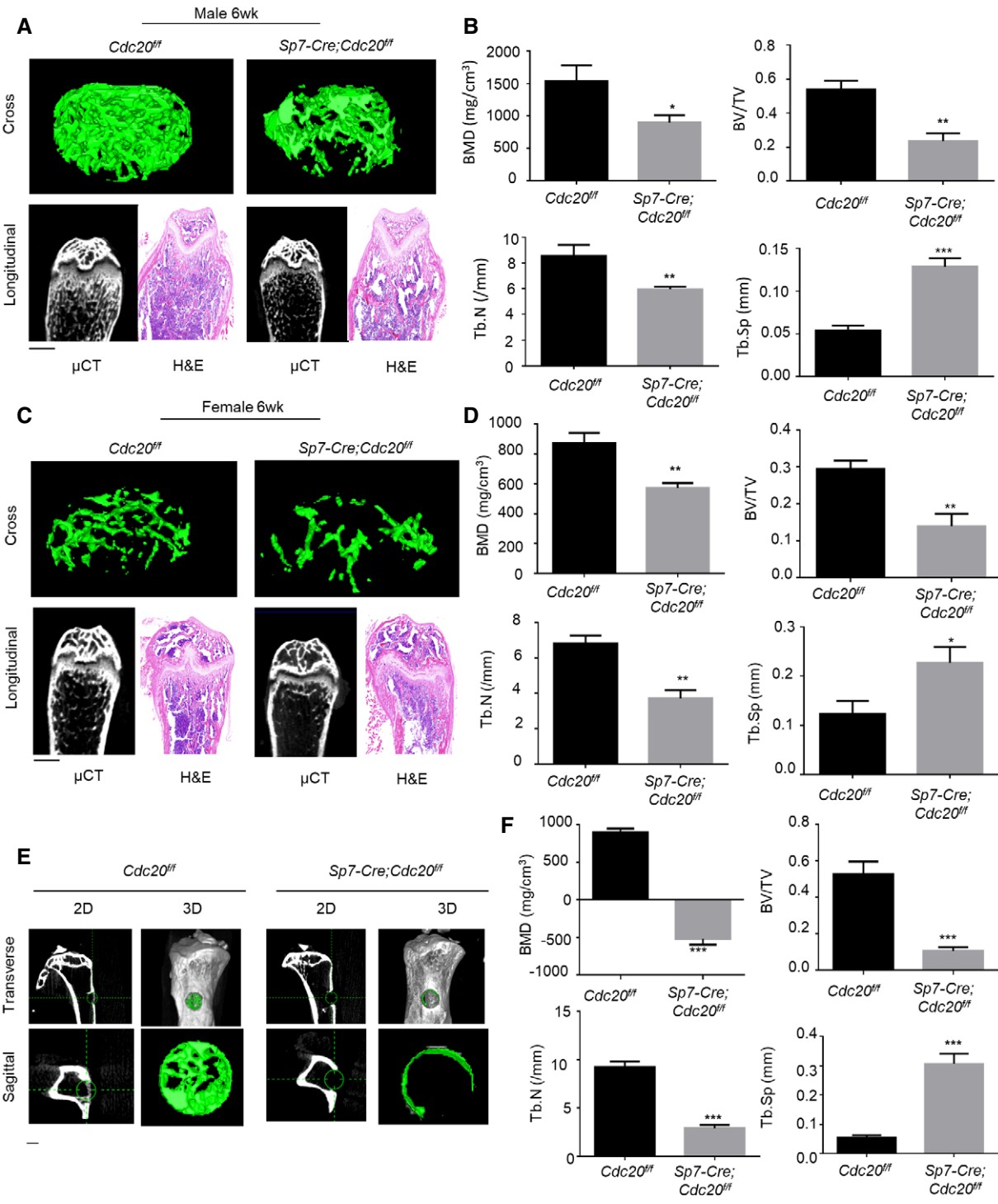

**Figure 1.**

osteoblast marker PINP and osteoclast marker CTX-1 of 6-week-old mice were measured through ELISA. The results showed that the bone formation biomarker PINP decreased in *Sp7-Cre; Cdc20^{f/f}* mice, while bone resorption marker CTX-1 did not change (Appendix Fig S3D and E). These results implied that the decreased function of osteoblasts was not due to the differences of cell numbers, while the function and numbers of osteoclasts remained stable.

Subsequently, we carried out skeletal defects through drilling holes in tibial cortical bone to evaluate the effect of Cdc20 on bone healing. The holes were drilled at the same side of the tibiae at about 2mm under the knees. These defects were conducted on 12-week-old mice and these mice were sacrificed 10 days later. Micro-CT analyses revealed that the cortical gaps in *Cdc20^{f/f}* control mice were almost bridged, while the holes in *Sp7-Cre; Cdc20^{f/f}* were filled

partially (Fig 1E). In accordance with that, the calculated mineralized callus of *Sp7-Cre; Cdc20^{f/f}* were significantly lower than the control ones (Fig 1F). These results unveiled that conditional knockout of Cdc20 caused reduced bone mass in skeleton and postponed bone healing process.

## CDC20 modulates osteogenic differentiation of BMSCs

To evaluate the potential role of CDC20 in the osteogenic differentiation of BMSCs, we explored the expression status of CDC20 in BMSCs after osteogenic induction. Our results showed that the CDC20 expression level was significantly elevated in human BMSCs (hBMSCs) during osteogenesis for 7 and 14 days, as determined by Western blot and qRT–PCR (Fig EV2A–D). To further investigate the important role of CDC20 in osteogenic differentiation of BMSCs, we gathered BMSCs from *Cdc20^{f/f}* control mice and *Sp7-Cre; Cdc20^{f/f}* experimental mice through flushing bone marrow and digesting bone tissues with collagenase. The knockout efficiency of Cdc20 was confirmed through Western blot and qRT–PCR analysis (Figs 2A and EV2E). Also, the expression of osteogenic marker Runt-related transcription factor 2 (*Runx2*) was greatly decreased in experimental groups (Fig EV2F). After cultured in osteogenic medium for 1 week, the *Sp7-Cre; Cdc20^{f/f}* BMSCs exhibited decreased osteogenic differentiation capability, as determined by ALP staining and quantification (Fig 2B and C). Next, we established CDC20 stable knockdown hBMSCs, and the transduction efficiency was measured by Western blot analyses, fluorescent staining, and qRT–PCR (Figs 2D and EV2G and H). After cultured in osteogenic medium for 7 days, the deficiency of CDC20 strongly inhibited the RUNX2 expression, as determined by qRT–PCR and Western blot assay (Fig EV2I and J). Moreover, the CDC20 knockdown hBMSCs showed decreased ALP staining and quantification (Fig 2E and F). Next, to figure out the functional domain of CDC20 regulating the osteogenic differentiation of BMSCs, we transfected truncated CDC20 plasmids encoding Myc-CDC20, Myc-CDC20 171–499 fragment containing WD40 domain (responsible for protein–protein interaction) or Myc-CDC20 1–170 fragment lacking WD40 domain into CDC20 knockdown cells (Figs 2G and EV2K and L). The ALP staining and quantification presented that the WD40 domain of CDC20 is essential for promoting the osteogenic differentiation of hBMSCs (Fig 2H and I). Furthermore, the stable CDC20 knockdown hBMSCs were loaded onto scaffolds and transplanted into the subcutaneous space of nude mice. After 8 weeks, the samples were harvested and subjected to histological analyses. H&E and Masson's trichrome showed that CDC20 knockdown hBMSCs generated less bone-like tissues when compared with control cells (Fig 2J and K). The expression of osteoblast markers osteocalcin (OCN) and osteopontin (OPN) decreased in CDC20sh scaffold using IHC staining (Appendix Fig S2F). These results showed that knockdown of CDC20 impaired osteogenic differentiation of BMSCs.

## CDC20 induces proteasome-dependent degradation of p65

To clarify the underlying mechanism of CDC20 regulating osteogenic differentiation, we investigated several key pathways and factors involved in regulating osteogenesis. Unexpectedly, we found that the expression of p65 target genes *IL-8, IL-6, ICAM1* were significantly increased in CDC20 knockdown cells after TNF-α stimulation

for 0.5 h (Fig EV3A–D). The expression of *IL-6* and *IL-8* gene increased, while *p65* gene did not change in CDC20si hBMSCs (Appendix Fig S4A). We next examined the effects of CDC20 on p65 stability by Western blot assay. In *Sp7-Cre; Cdc20^{f/f}* BMSCs, the expression level of p65 was largely improved compared to *Cdc20^{f/f}* BMSCs (Fig 3A). In CDC20si hBMSCs, there was an obvious increase in p65 expression level (Fig 3B). Moreover, we got the same results in CDC20 knockdown HEK293T cells (Fig EV3E). These data confirmed that the regulation of p65 by CDC20 was a general phenomenon. Furthermore, Western blot analyses showed that knockdown of CDC20 led to increased p65 level in the nucleus after the inflammatory factor TNF-α stimulation (Fig 3C and D). In addition, Fig 3E and F showed that overexpression of CDC20 decreased the nuclear protein level of p65. Overexpression of CDC20 significantly reduced the expression of p65, while treatment with MG132 (the proteasome inhibitor) blocked the CDC20 triggered p65 decrease in hBMSCs (Fig 3G), which demonstrated that CDC20 degrades p65 through the proteasome-dependent pathway. Overexpression of CDC20 reduced p65 protein level in HEK293T cells were also shown in Fig EV3F. In addition, we examined the expression levels of other NF-κB pathway proteins, the results showed increased expression of Phospho-NF-κB p65 (Ser536) (p-p65) together with p65 in CDC20sh cells (Appendix Fig S4B). No interaction was found between IκBα and CDC20 (Appendix Fig S4C and D). In addition, we investigated the relative expressions of CDC20, p65 and osteoblast markers (OCN and RUNX2) of *Cdc20^{f/f}* and *Sp7-Cre; Cdc20^{f/f}* bone sections using immunofluorescence (IF). In 6-week-old conditional knockout mice femurs, the expression of CDC20, OCN, and RUNX2 decreased, while p65 expression increased (Appendix Fig S3F), which were correlated with the cellular results above. Concerning about the chondrocytes, the expression of the specific marker, collagen type II (COL2) was evaluated through immunofluorescence. The results showed that the expressions of COL2 and CDC20 were stable in *Cdc20^{f/f}* and *Sp7-Cre; Cdc20^{f/f}* cartilage of 6-week-old mice femurs (Appendix Fig S4F). Altogether, these results suggested that CDC20 inhibited NF-κB signaling by inducing p65 proteasome-dependent degradation.

## CDC20 interacts with p65

We next investigated whether CDC20 interacts with p65 and then modulates its ubiquitination and degradation. To this end, we co-transfected Myc-tagged CDC20 and Flag-tagged p65 into HEK293T cells. Lysates of co-transfected cells were subjected to co-immunoprecipitation (Co-IP) assay with either anti-Myc or anti-Flag followed by probing with anti-Flag or anti-Myc, respectively. The results clearly demonstrated an interaction between the exogenous Myc-CDC20 and Flag-p65 proteins (Fig 4A and B). Next, we found Myc-tagged CDC20 interacted with endogenous p65 and Flag-tagged p65 bounded with endogenous CDC20 (Fig 4C and D). Furthermore, the whole-cell extracts of HEK293T cells were prepared and subjected to a Co-IP assay with anti-CDC20 or anti-p65 and then probed with anti-p65 or anti-CDC20, respectively (Fig 4E and F). Also, the interaction between CDC20 and p65 was obviously detected in hBMSCs (Fig EV3G and H), and similar results were obtained in mouse BMSCs (mBMSCs) (Fig EV3I and J). Using immunofluorescence, the co-localization of CDC20 and p65 were found in hBMSCs (Fig EV3K) and 6-week-old mice bone sections

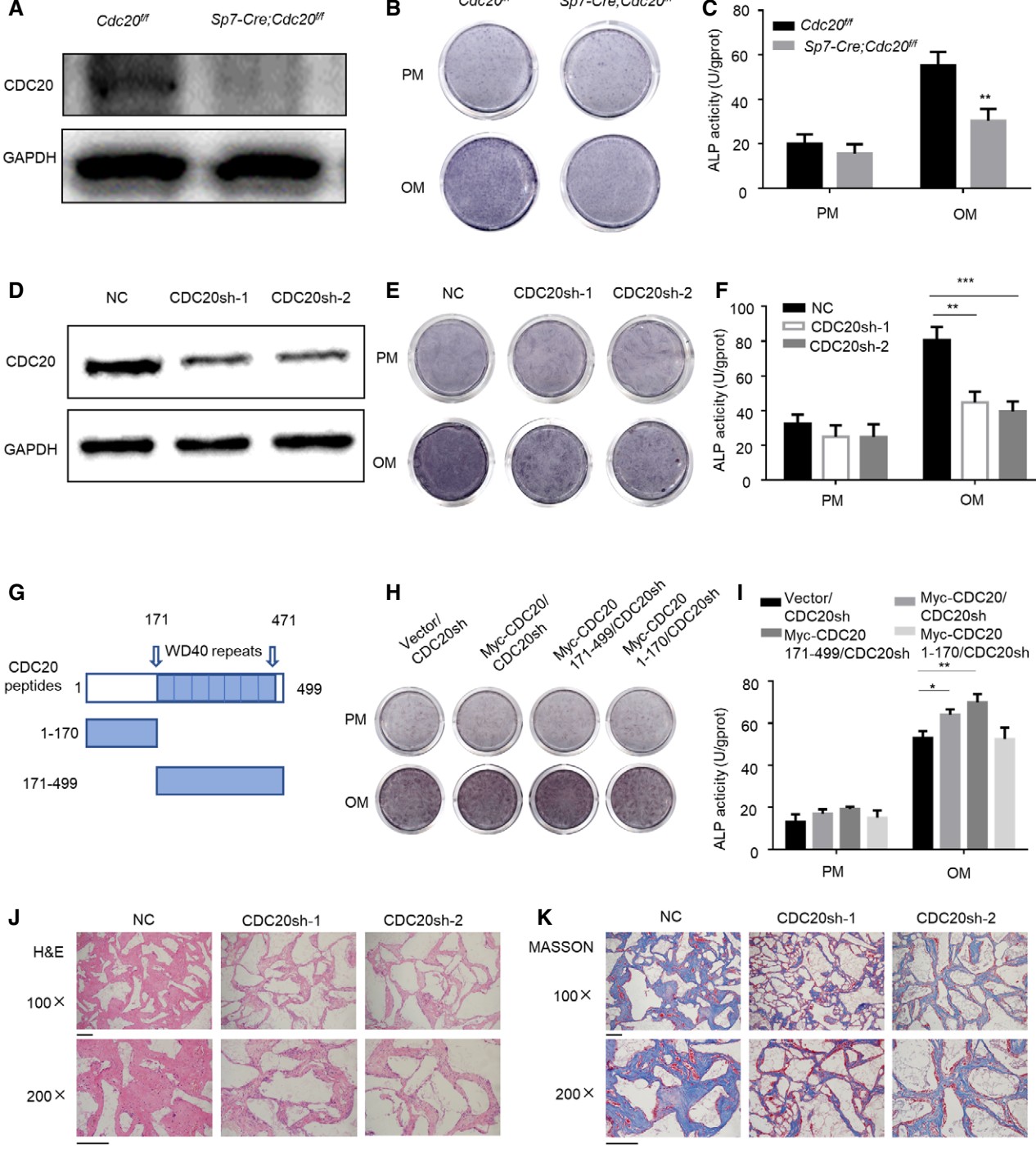

**Figure 2. CDC20 modulates osteogenic differentiation of BMSCs.**

A   Western blot analyses of the knockout efficiency of Cdc20 in $Cdc20^{f/f}$ and $Sp7-Cre;Cdc20^{f/f}$ BMSCs.

B, C   The ALP staining (B) and ALP quantification (C) of $Cdc20^{f/f}$ and $Sp7-Cre;Cdc20^{f/f}$ BMSCs after 7 days osteogenic differentiation ($n = 6$).

D   The efficiency of CDC20 knockdown determined by Western blot in hBMSCs.

E, F   The ALP staining (E) and ALP quantification (F) of NC and CDC20sh hBMSCs after 7 days osteogenic differentiation ($n = 5$).

G   Schematic representation of the CDC20 truncated mutant fragments. The boundaries of its WD40 domain (amino acids 171–471) were shown.

H, I   The ALP staining (H) and ALP quantification (I) of Vector and truncated fragments overexpression of CDC20 in CDC20sh hBMSCs after 7 days osteogenic differentiation ($n = 5$).

J, K   The H&E staining (J) and Masson's trichrome (K) of the histological sections from implanted NC and CDC20sh hBMSCs scaffold hybrids. Scale bar, 100 μm.

Data information: Data are displayed as mean ± SD and show one representative of $n \geq 3$ independent experiments with three biological replicates. Statistical significance was calculated by one-way ANOVA followed by a Tukey's post hoc test and defined as *$P < 0.05$; **$P < 0.01$; ***$P < 0.001$.

(Appendix Fig S4E). To detect whether CDC20 directly interacts with p65, glutathione S-transferase (GST) pull-down experiments were performed. The purified recombinant proteins GST-CDC20 or GST-p65 expressed in bacterial were subjected to pull-down assay with anti-GST followed by analyzing with anti-p65 and anti-CDC20, separately. Figure 4G and H exhibited that CDC20 interacted with p65 directly. These results showed the strong interaction between CDC20 and p65.

### The functional domain of CDC20 and p65

We further investigated the functional domain of CDC20 and p65 for interaction. Previous studies have shown that the WD40 domain of CDC20 is important for protein–protein interaction, so it was of interest to determine whether WD40 domain could mediate the binding between CDC20 and p65. We transfected HEK293T cells with Myc-tagged CDC20 fragments as mentioned above, the

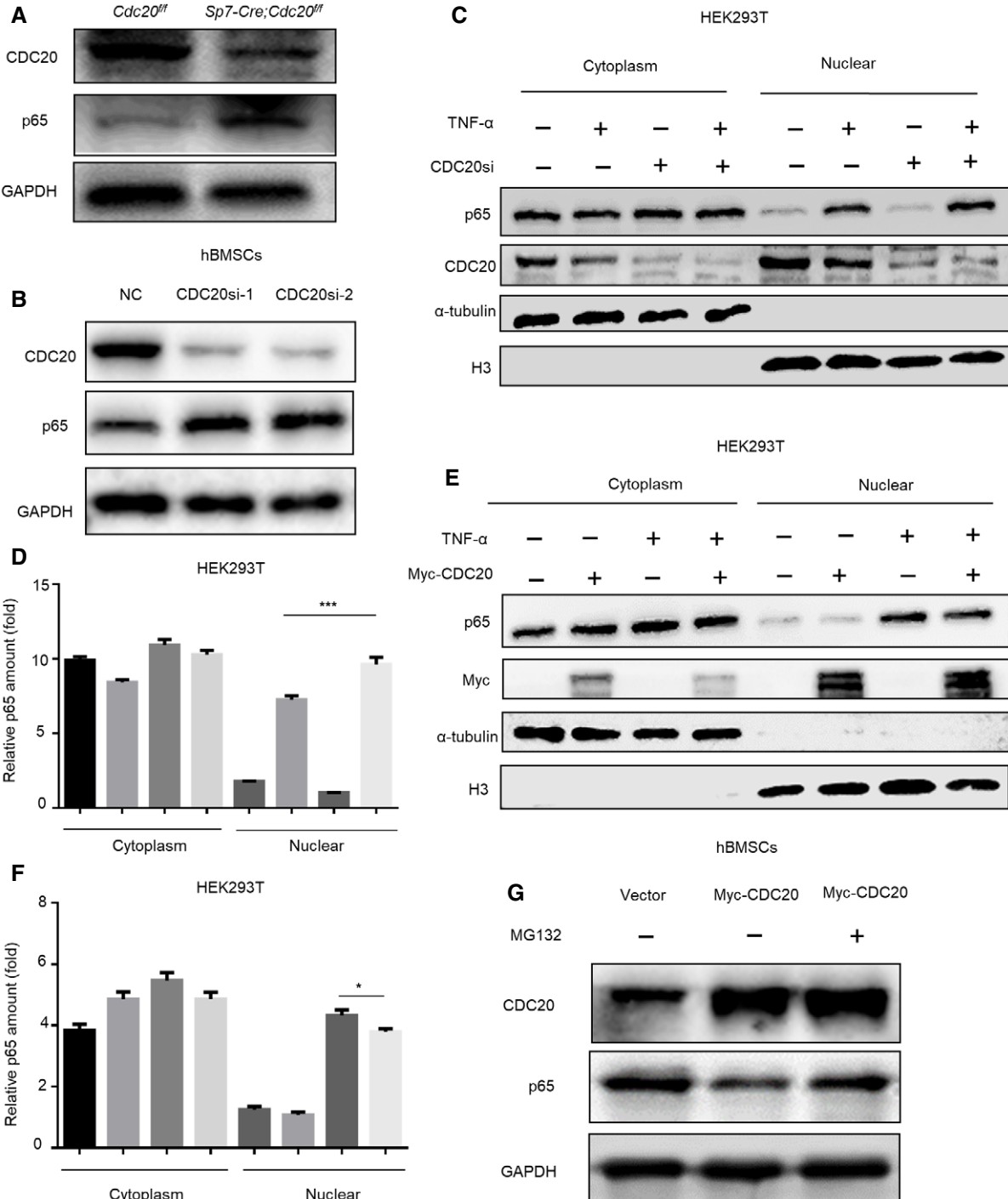

**Figure 3.**

**Figure 3.  CDC20 induces proteasome-dependent degradation of p65.**

A  Western blot analyses of the degradation of endogenous p65 protein in BMSCs from *Sp7-Cre;Cdc20^f/f^* and *Cdc20^f/f^* mice.
B  Western blot analyses of the degradation of endogenous p65 protein in NC and CDC20si hBMSCs.
C  Western blot analyses of the degradation of nuclear p65 protein after the knockdown of CDC20. The cytoplasmic and nuclear protein of NC and CDC20si HEK293T cells were extracted after TNF-α treatment for 30 min.
D  The graphically presented amount of p65 of NC and CDC20si HEK293T cells was normalized with α-tublin and H3 expression. The percent of p65 protein was calculated accordingly ($n = 6$).
E  Western blot analyses of the degradation of nuclear p65 protein after the overexpression of CDC20. The cytoplasmic and nuclear protein of overexpressed Myc-CDC20 and vector HEK293T cells were extracted after TNF-α treatment for 30 min.
F  The graphically presented amount of p65 of overexpressed Myc-CDC20 and vector HEK293T cells was normalized with α-tublin and H3 expression. The percent of p65 protein was calculated accordingly ($n = 5$).
G  Western blot analyses of the degradation of p65 protein after the overexpression of Myc-CDC20. hBMSCs were transfected with vector and Myc-CDC20 plasmids for 36 h, cells were treated with or without 10 μM MG132 for 6 h before collected.

Data information: Data are displayed as mean ± SD and show one representative of $n \geq 3$ independent experiments with three biological replicates. Statistical significance was calculated by one-way ANOVA followed by a Tukey's post hoc test and defined as *$P < 0.05$; ***$P < 0.001$.

schematic illustration of WD40 domain was as indicated in Fig 2G and the expression of each fragment was shown in Fig EV2K. Whole-cell extracts were lysed and subjected to immunoprecipitation with Myc antibody and analyzed by Western blot with p65 antibody. Our results showed that p65 specifically interacted with the CDC20 fragment containing WD40 repeats domain (Fig 5A and B). We further performed Co-IP experiments using Flag-p65 fragments to identify the region involved in the association with CDC20. p65 contains a 300 amino acid conserved region: the Rel homology domain (RHD), which is the DNA-binding domain. We first constructed the separate plasmids; the structure of RHD domain and Western blot analyses of each fragment are shown in Fig 5C and D. Then, we transfected HEK293T cells with plasmids encoding Flag-p65 1–310 fragment (containing RHD domain) or Flag-p65 311–551 fragment lacking RHD domain and performed Co-IP assay with anti-CDC20 antibodies. Results showed that only the exogenous expressed Flag-p65 containing RHD domain interacted with endogenous CDC20 (Fig 5E and F). To further clarify the functional domain of p65 and CDC20 for interaction, cell extracts expressing Myc-CDC20 171–499 fragment and Flag-p65 1–310 fragment were subjected to immunoprecipitation for Myc or Flag, respectively. Figure 5G and H confirmed the interaction between the indicated functional domains of p65 and CDC20 separately. Overall, the WD40 domain, through which CDC20 recruits substrates to the APC/C complex (Yu, 2007), is responsible for binding the RHD domain of p65.

## CDC20 regulates p65 degradation in an APC11-dependent manner

CDC20 functions as one of the imporC20 regulates p65 degradation in an tant substrate receptors among APC/C complex, we then explored whether APC/C is the E3 enzyme responsible for the degradation of p65. The catalytic module, RING domain subunit APC11, and the Cullin domain subunit APC2 were examined in the degradation assay. Co-IP assay detected that p65 only binds to APC11 and has no interaction with APC2 (Figs 6A and B, and EV4A and B). Moreover, the expression level of p65 was increased in APC11 knockdown cells (Fig 6C). In contrast, overexpression of APC11 led to a sharp decrease of p65, while treatment with MG132 blocked the APC11-induced reduction of p65 expression (Fig 6D). Furthermore, knockdown or overexpression of APC2 did not affect the expression

of p65 (Fig EV4C and D). These results suggested that APC11 might be the E3 ligase responsible for p65 degradation. Since CDC20 is the co-activator of APC/C complex, we wondered whether CDC20 plays a role in the interaction between APC11 and p65. Our results showed that knockdown of CDC20 inhibited the binding between APC11 and p65 (Fig 6E). In addition, CDC20 deficiency could rescue the degradation of p65 triggered by APC11 (Fig 6F). These results suggested that CDC20 regulates p65 degradation in an APC11-dependent manner. As another member of co-activator of APC/C complex, we also detected the effect of CDH1 in the interaction of APC11 and p65 degradation. Knockdown of CDH1 did not influence the expression level of p65 and CDC20 (Fig EV4E and F). Also, the interaction between APC11 and p65 did not change in CDH1sh cells (Fig EV4G). We then examined the potential effect of APC11 in the regulation of osteogenic differentiation of BMSCs. We used siRNA to construct APC11 knockdown hBMSCs, the knockdown efficiency was determined by Western blot and qRT–PCR analyses (Figs 6G and EV4H). ALP staining and quantification showed that knockdown of APC11 significantly impaired the osteogenic differentiation of hBMSCs (Fig 6H and I). These results showed that CDC20 regulates p65 degradation is dependent of APC11.

## The ubiquitination of p65 by APC11^CDC20

As indicated above, CDC20 promotes p65 degradation in an APC11-dependent manner. We next examined whether APC11^CDC20 ubiquitinates p65. HEK293T cells were co-transfected with Flag-p65, HA-Ubiquitin, with or without Myc-CDC20. Immunoprecipitation assay showed the high mass ladder of ubiquitin only existed in the presence of CDC20, showing that CDC20 strengthened the ubiquitination of p65 (Fig 7A). Also, using similar methods, HEK293T cells were co-transfected with Flag-p65, His-Ubiquitin, with or without HA-APC11. Ubiquitinated proteins were captured by magnetic beads and detected by anti-His immunoblot. Figure 7B showed that overexpression of APC11 in HEK293T cells significantly induced the ubiquitination of p65. Additionally, we tried to figure out whether APC11^CDC20 directly ubiquitinates p65. An *in vitro* ubiquitination assay was performed using purified proteins, including HA-ubiquitin, ubiquitin-activating E1 enzyme UBE1, ubiquitin-conjugating E2 enzyme Ubch10, Flag-p65 purified from cell extracts, together with GST, GST-CDC20, or GST-APC11 purified from bacterial. Notably, significant ubiquitination of p65 was detected with GST-CDC20 or

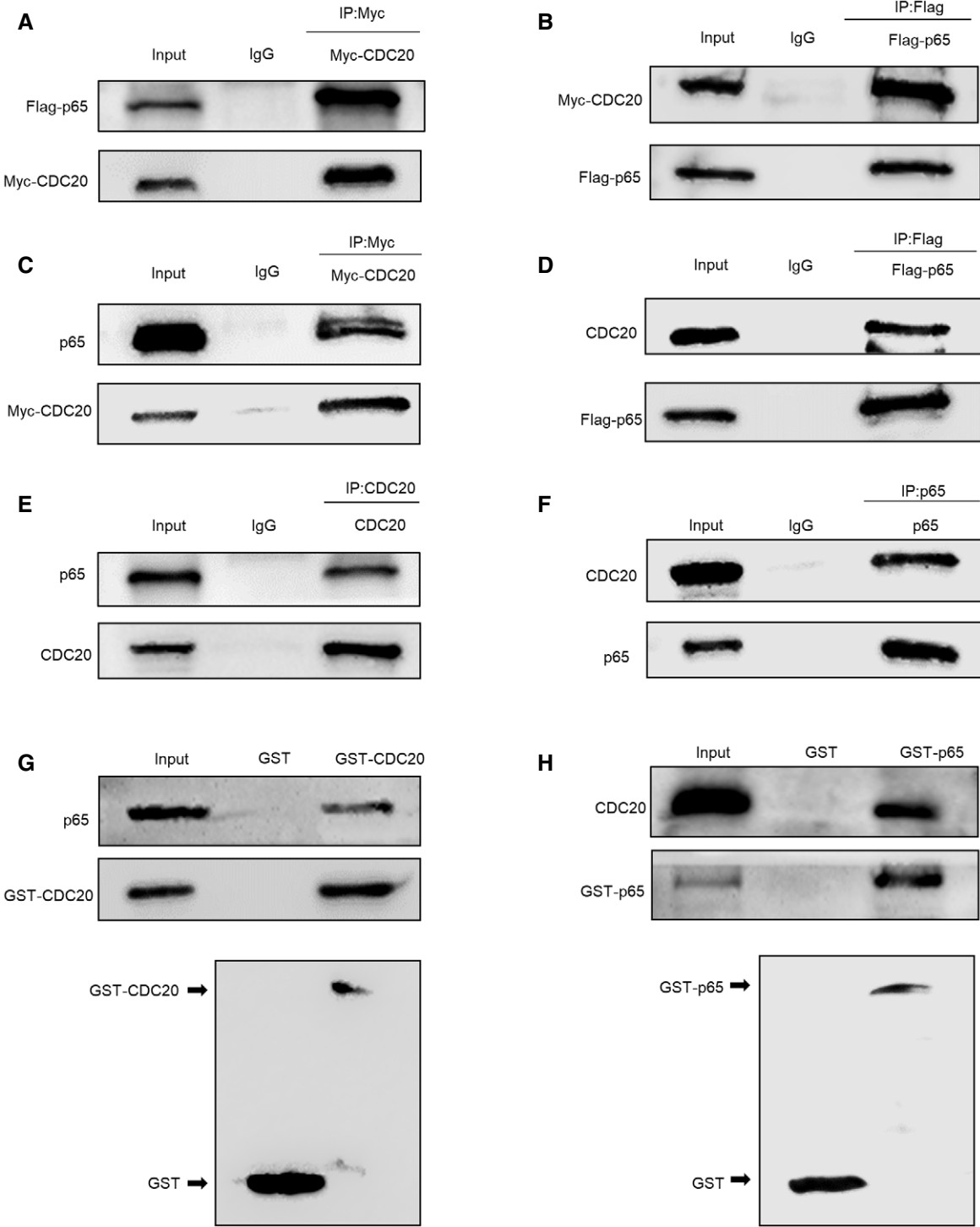

**Figure 4. CDC20 interacts with p65.**

A, B   Co-immunoprecipitation of ectopically expressed Myc-CDC20 with ectopically expressed Flag-p65 in HEK293T cells.

C, D   Co-immunoprecipitation of ectopically expressed Myc-CDC20 with endogenous p65 and ectopically expressed Flag-p65 with endogenous CDC20 in HEK293T cells.

E, F   Co-immunoprecipitation of endogenous CDC20 with endogenous p65 in HEK293T cells.

G       Co-immunoprecipitation of purified GST-CDC20 protein with endogenous p65 in HEK293T cells and the construct of GST-CDC20. Whole-cell lysates (WCL) were subjected to perform GST pull-down analyses with purified GST-CDC20 and GST as a negative control.

H       Co-immunoprecipitation of purified GST-p65 protein with endogenous CDC20 in HEK293T cells and the construct of GST-p65. Whole-cell lysates (WCL) were subjected to perform GST pull-down analyses with purified GST-p65 and GST as a negative control.

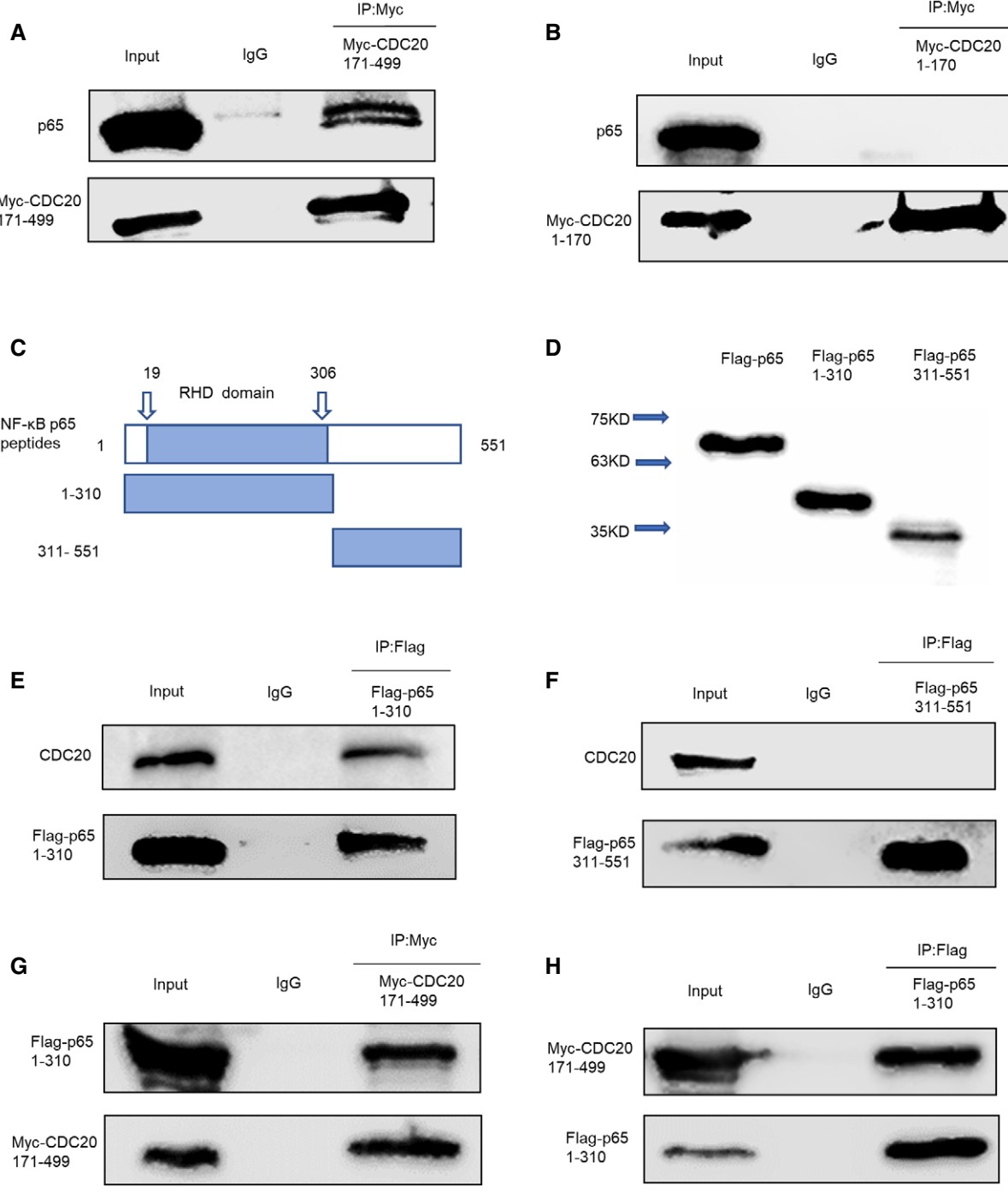

**Figure 5. The functional domain of CDC20 and p65.**

A, B    Co-immunoprecipitation of ectopically expressed Myc-CDC20 171–499 fragment (containing WD40 domain) (A) rather than Myc-CDC20 1–170 fragment (lacking WD40 domain) (B) with endogenous p65 in HEK293T cells.

C    Schematic representation of RHD domain location in p65 peptides (amino acids 19-306).

D    Western blot analyses of Flag-p65, Flag-p65 1–310 fragment (containing RHD domain), Flag-p65 311–551 fragment (lacking RHD domain) plasmids expression in HEK293T cells.

E, F    Co-immunoprecipitation of ectopically expressed Flag-p65 1–310 fragment (containing RHD domain) (E) rather than Flag-p65 311–551 fragment (lacking RHD domain) (F) with endogenous CDC20 in HEK293T cells.

G, H    Co-immunoprecipitation of ectopically expressed Myc-CDC20 171–499 with ectopically expressed Flag-p65 1–310 fragment in HEK293T cells.

GST-APC11 (Fig 7C and D). The combination of GST-APC11 and GST-CDC20 exerted more p65 ubiquitination than GST-CDC20 (Appendix Fig S5A). These results support p65 as a direct APC11$^{CDC20}$ substrate. Furthermore, in CDC20sh or APC11sh cells, the ubiquitinated p65 were far less than the control ones (Appendix Fig S5B and C), indicating that CDC20, as well as APC11 played an important role in the ubiquitination of p65. Similarly, *Sp7-Cre; Cdc20$^{f/f}$* BMSCs showed significantly decreased ubiquitination compared to *Cdc20$^{f/f}$* controls obtained from 6-week-old mice (Appendix Fig S5D). The studies above showed that the WD40

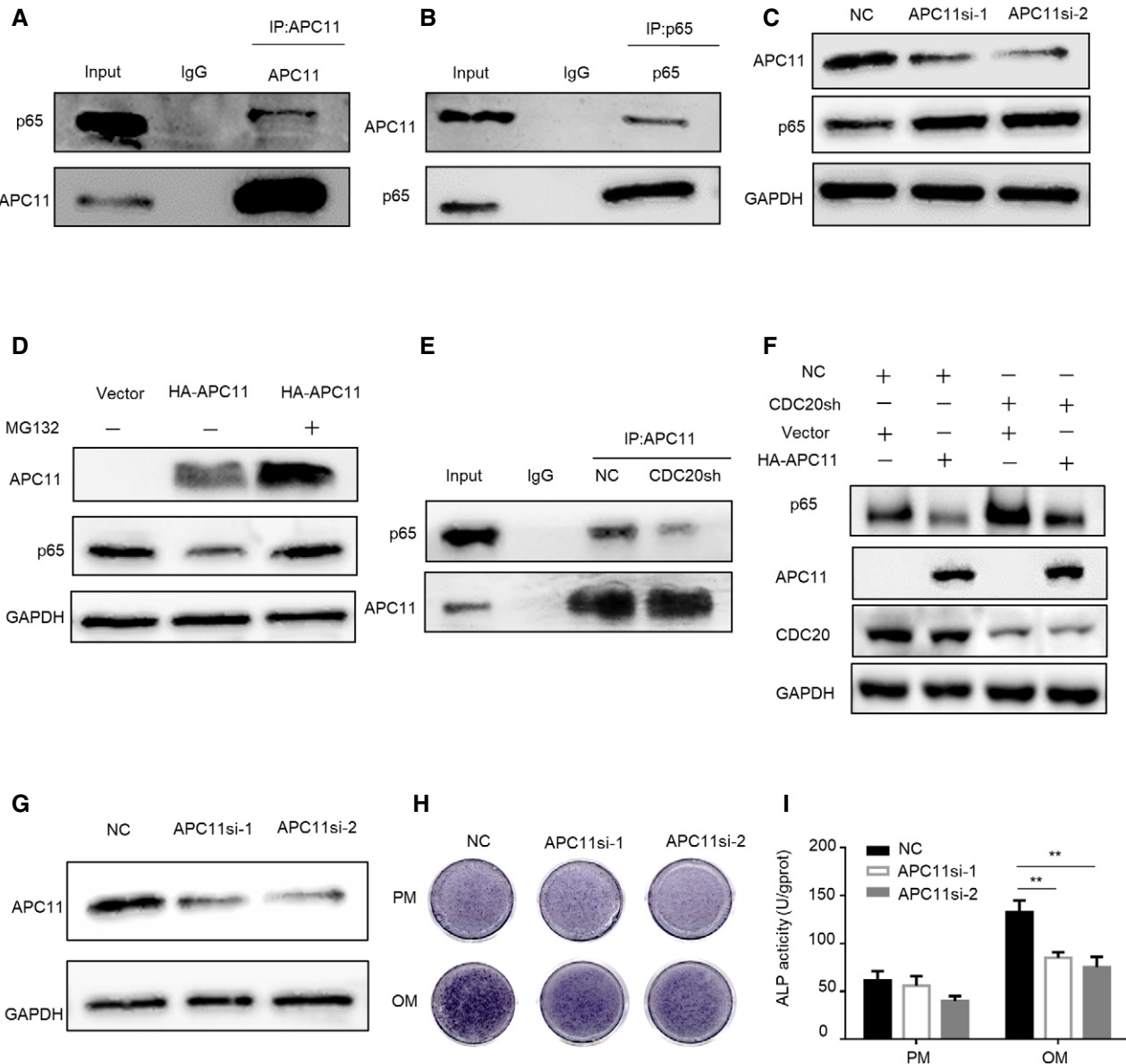

**Figure 6.  CDC20 regulates p65 degradation in an APC11-dependent manner.**

A, B   Co-immunoprecipitation of endogenous APC11 with endogenous p65 in HEK293T cells.
C       Western blot analyses of the degradation of p65 protein in NC and APC11si HEK293T cells.
D       Western blot analyses of the degradation of p65 protein by APC11. HEK293T cells were transfected with vector and HA-APC11 plasmids for 36 h, cells were treated with or without 10 μM MG132 for 6 h before collection.
E       Co-immunoprecipitation of APC11 with endogenous p65 in NC and CDC20sh HEK293T cells.
F       Western blot analyses of p65 expression in NC or CDC20sh HEK293T cells after the overexpression of Vector or HA-APC11 plasmids.
G       The efficiency of APC11 knockdown determined by western blot in hBMSCs.
H, I    The ALP staining (H) and ALP quantification (I) of NC and APC11si hBMSCs after 7 days osteogenic differentiation (*n* = 6).

Data information: Data are displayed as mean ± SD and show one representative of *n* ≥ 3 independent experiments with three biological replicates. Statistical significance was calculated by one-way ANOVA followed by a Tukey's post hoc test and defined as **$P$ < 0.01.
Source data are available online for this figure.

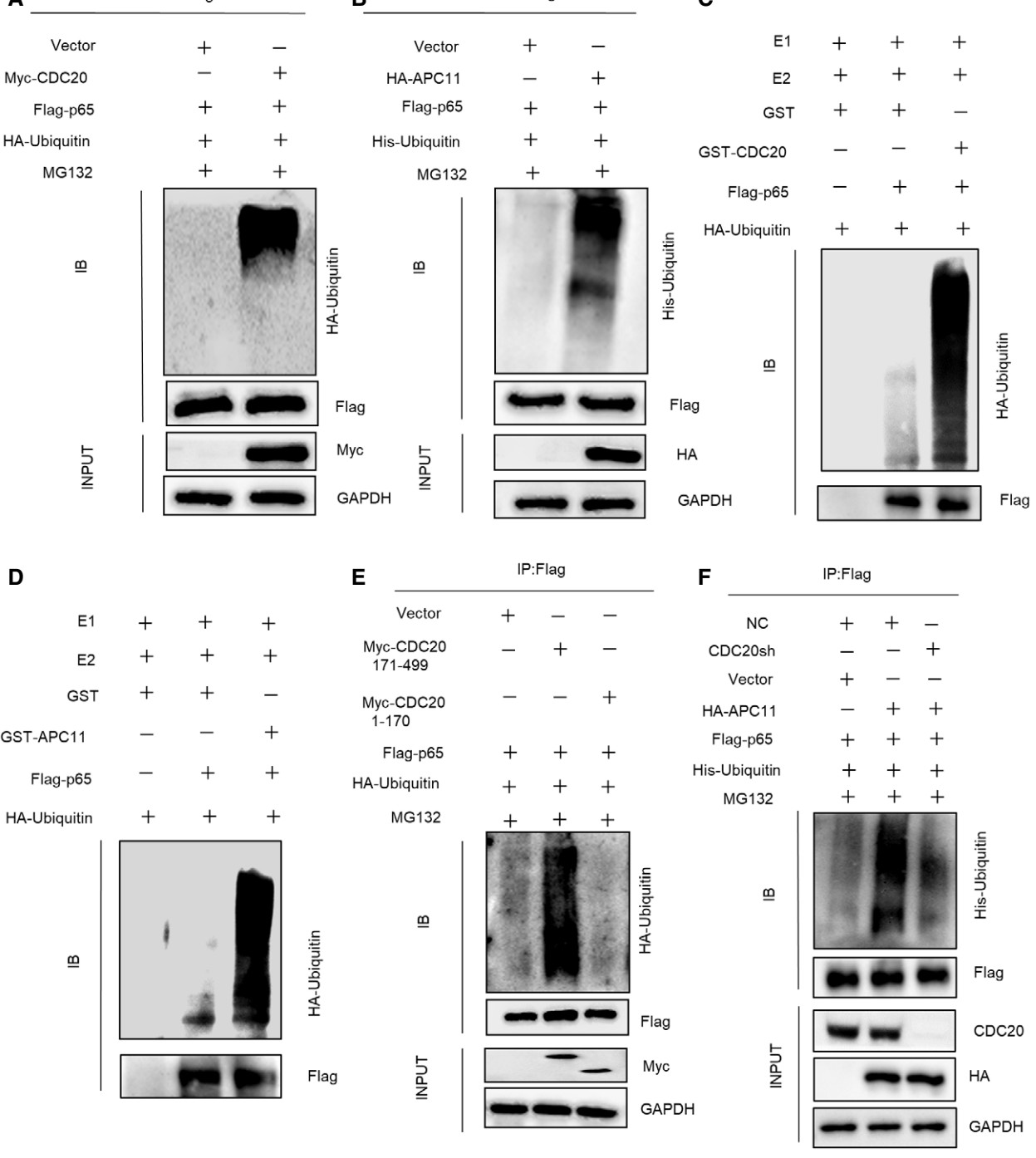

**Figure 7. The ubiquitination of p65 by APC11$^{CDC20}$.**

A   Immunoblot of Flag-p65-linked ubiquitination promoted by Myc-CDC20. HEK293T cells were co-transfected with Flag-p65, HA-Ubiquitin, with or without Myc-CDC20 or vector plasmids for 36 h. Transfected cells were then treated with 10 μM MG132 for 6 h before collection.

B   Immunoblot of Flag-p65-linked ubiquitination promoted by HA-APC11. HEK293T cells were transfected with Flag-p65, His-Ubiquitin, with or without HA-APC11 or Vector plasmids for 36 h. Transfected cells were then treated with 10 μM MG132 for 6 h before collection.

C, D   Immunoblot of Flag-p65-linked *in vitro* ubiquitination promoted by GST-CDC20 (C) or GST-APC11 (D). Bacterially expressed and purified GST-CDC20 or GST-APC11 were incubated with purified proteins, including HA-Ubiquitin, E1, E2, and Flag-p65 as indicated at 32°C for 1 h.

E   Immunoblot of Flag-p65-linked ubiquitination promoted by truncated Myc-CDC20 fragments. HEK293T cells were transfected with Flag-p65, HA-Ubiquitin, with or without Myc-CDC20 171–499 fragment (containing WD40 domain), Myc-CDC20 1–170 fragment (WD40 domain deficient), and vector plasmids. Transfected cells were then treated with 10 μM MG132 for 6 h before collection.

F   Immunoblot of Flag-p65-linked ubiquitination promoted by HA-APC11 in NC and CDC20sh cells. NC or CDC20sh HEK293T cells were co-transfected with Flag-p65, His-Ubiquitin, with or without HA-APC11 and vector plasmids. Transfected cells were treated with 10 μM MG132 for 6 h before collection.

domain of CDC20 interacted with p65. Thus, to determine whether the functional domain of CDC20 is indispensable for the ubiquitination of p65, CDC20 fragments (Fig 2G), Flag-p65, and HA-Ubiquitin were separately transfected into HEK293T cells. Results showed that CDC20 lacking WD40 domain largely impaired the ubiquitination of p65 (Fig 7E). Most importantly, CDC20 deficiency impaired

APC11-triggered p65 ubiquitination (Fig 7F), while CDH1 deficiency exerted no effort in p65 ubiquitination (Appendix Fig S5E). We further investigated whether APC2 is indispensable in p65 ubiquitination catalyzed by APC11. We performed the ubiquitination assay in NC and APC2sh cells, with the overexpression of APC11, and found no significant difference of the ubiquitination of p65

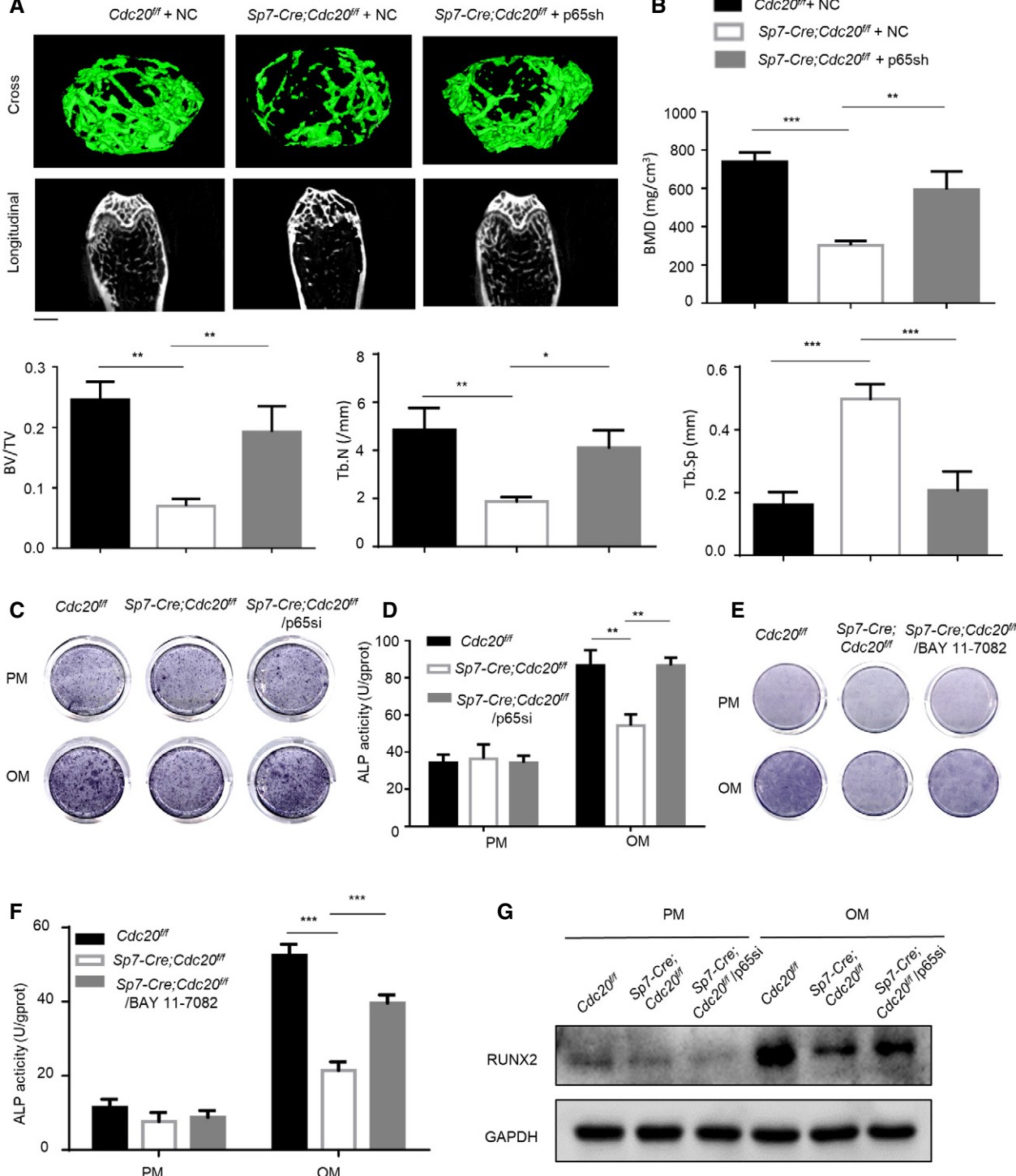

Figure 8.

**Figure 8. CDC20 regulated osteogenic differentiation of BMSCs in a p65-dependent manner.**

A  Representative micro-CT images of trabecular bone from the femoral metaphysis of *Cdc20^f/f* with negative control lentivirus, *Sp7-Cre;Cdc20^f/f* with negative control lentivirus, and *Sp7-Cre;Cdc20^f/f* with p65sh lentivirus. Scale bar, 500 μm.

B  Histomorphometric analyses of *Cdc20^f/f* with negative control lentivirus, *Sp7-Cre;Cdc20^f/f* with negative control lentivirus, and *Sp7-Cre;Cdc20^f/f* with p65sh lentivirus (*n* = 6).

C, D  The ALP staining (C) and ALP quantification (D) of BMSCs from *Cdc20^f/f* mice and *Sp7-Cre;Cdc20^f/f* mice after 7 days osteogenic differentiation treated with negative control or p65si (*n* = 5).

E, F  The ALP staining (E) and ALP quantification (F) of BMSCs from *Cdc20^f/f* and *Sp7-Cre;Cdc20^f/f* mice after 7 days osteogenic differentiation treated with BAY 11-7082 (NF-κB pathway inhibitor) (*n* = 5).

G  The expression of RUNX2 in BMSCs from *Cdc20^f/f* mice and *Sp7-Cre;Cdc20^f/f* mice after 7 days osteogenic differentiation treated with negative control or p65si by Western blot analyses.

Data information: Data are displayed as mean ± SD and show one representative of *n* ≥ 3 independent experiments with three biological replicates. Statistical significance was calculated by one-way ANOVA followed by a Tukey's post hoc test and defined as *$P < 0.05$; **$P < 0.01$; ***$P < 0.001$.

(Appendix Fig S6A). The substrate cyclin B1 was measured to demonstrate the effective depletion of APC2, APC11, and CDH1 (Appendix Fig S6B–D), and our results implied that APC2 did not influence the ubiquitination of p65 by APC11. These results suggested that CDC20 is a co-activator for APC11-mediated ubiquitination of p65.

## CDC20 regulated osteogenic differentiation of BMSCs in a p65-dependent manner

In order to figure out the functional connection between CDC20 and p65 in the osteogenic differentiation process, we injected lentivirus expressing p65 shRNA or negative control into Cdc20 conditional knockout mice from the tail intravenously. We injected $8 \times 10^8$ TU/ml lentiviruses 100 μl every 2 weeks for twice via the tail vein. Four weeks later, the mice were sacrificed. We captured the pictures of the gross appearance of NC and p65 knockdown mice and compared the body weight. The results showed that the NC and p65sh mice were healthy, and there were no differences in the gross appearances and body weights (Appendix Fig S7A and B). The Western blot assay affirmed the knockdown of p65 in the liver sections (Appendix Fig S7C). Then, we conducted H&E staining and TUNEL staining of the liver; no discernable differences were found, suggesting that apoptosis was not induced in p65sh mice (Appendix Fig S7D). To clarify the efficiency of p65 knockdown in bone sections, lentivirus was distributed in mice (Fig EV5A), the expression levels of p65 in BMSCs obtained from femurs and tibiae were examined according to qRT–PCR and Western blot analyses (Fig EV5B and C). Micro-CT analyses of the distal femur metaphysis showed an impairment in trabecular bone micro-architecture in *Sp7-Cre; Cdc20^f/f* mice, and the bone loss was effectively blocked treated with p65sh lentivirus for 4 weeks (Fig 8A). Quantitative measurements indicated that the area of bone formation of the Cdc20 conditional knockout group was significantly decreased compared with the negative control group and Cdc20/p65sh group (Fig 8B). The BMD and BV/TV parameters in *Sp7-Cre; Cdc20^f/f* treated with negative control lentivirus were greatly lower than *Sp7-Cre; Cdc20^f/f* treated with p65sh lentivirus and their *Cdc20^f/f* littermates. *Sp7-Cre; Cdc20^f/f* mice with negative control lentivirus also had reduced Tb.N and increased Tb. Sp.

Furthermore, we generated siRNA to silence p65 in mBMSCs obtained from mice. After the osteogenic induction for 7 days, the reduced osteogenic differentiation ability by Cdc20 knockdown was rescued in Cdc20 and p65 double knockdown cells, as indicated by ALP staining and ALP quantification (Fig 8C and D). Similar results

were confirmed in hBMSCs (Fig EV5D and E). Furthermore, we found that the treatment of BAY11-7082 (the NF-κB pathway inhibitor) could strongly rescue ALP activity in *Sp7-Cre; Cdc20^f/f* BMSCs (Fig 8E and F). Similar results were obtained in CDC20 knockdown hBMSCs (Fig EV5F and G), which demonstrated that APC11^CDC20 modulates osteogenic differentiation of BMSCs through NF-κB signaling pathway. In addition, after osteogenic induction for 7 days, the decreased expression of osteogenic marker RUNX2 caused by Cdc20 deficiency was blocked by inhibition of p65 determined by qRT-PCR and Western blot (Figs 8G and EV5H). As a whole, these results illustrated that the role of CDC20 in bone regulation was dependent on p65. The schematic presentation of the whole results was presented in Appendix Fig S7E. The statistical analyses of significantly different Western blot lanes are presented in Appendix Fig S8A–Q.

# Discussion

Our present study unraveled that CDC20, one of the two co-activators of APC/C, interacts and degrades p65, thus represses NF-κB downstream transcription to manipulate bone turnover. In previous studies, cyclin A2 and cyclin B1 are degraded in precise temporal manner by APC/C^CDC20 in G1 phase during mitosis process (Clute & Pines, 1999; Elzen & Pines, 2001). In addition, APC/C^CDC20 mediates degradation of Nek2A and cyclin A prior to the metaphase–anaphase transition (Geley *et al*, 2001; Hames, 2001). Except for cell cycle regulation, APC/C^CDC20 regulates presynaptic differentiation and dendrite morphogenesis through degradation of the transcription factors of NeuroD2 (Yang *et al*, 2009) and Id1 (Kim *et al*, 2009). Our research confirmed a strong interaction between CDC20 and p65 and figured out that p65 was a novel substrate of APC/C^CDC20. In terms of the functional domain, CDC20 and CDH1 contain WD40 repeats which bind directly to APC/C targets for ubiquitination and may serve as a bridge between substrates and enzymes (Burton & Solomon, 2001; Burton *et al*, 2005). Previous studies have reported that APC/C^CDC20 governed apoptosis by interacting and degrading Bim through its C-terminal WD40 repeats motif (Wan *et al*, 2014). In accordance with this study, we confirmed that APC/C^CDC20 regulated osteogenesis of BMSCs via interacting and degrading p65 through its C-terminal WD40 domain.

The APC/C displays substrates specificities depending on its association with the co-activator, CDC20 or CDH1 (Pines, 2011). CDC20 binds and activates APC/C during mitosis, while CDH1 binds and activates APC/C in mitosis and during G1 (Schwab *et al*, 1997;

Visintin, 1997). CDH1 has vital functions in controlling genomic and cell cycle progression by targeting its downstream substrates for ubiquitination and destruction (Harper *et al*, 2002; Peters, 2006). Besides its roles in cell cycle regulation, CDH1 has been found to play critical roles in various cellular processes including DNA cellular metabolism (Tudzarova *et al*, 2011), damage repair (Bassermann *et al*, 2008; Gao *et al*, 2009), neuronal development (Lasorella *et al*, 2006; Stegmüller *et al*, 2006), and cell migration (Naoe *et al*, 2010). Moreover, recent studies have revealed an APC/C-independent role of CDH1 in regulating osteoblast differentiation through disrupting the inter-molecular interaction of Smurf1 dimers, contributing to the activation of E3 ligase Smurf1 (Wan *et al*, 2011). The effect of CDC20, another co-activator of APC/C, on regulating bone formation has not been reported previously. Our study demonstrated the APC/C-dependent role of CDC20 in inducing osteogenesis of BMSCs. Among the APC/C complex, the catalytic core, a heterodimeric unit of APC11 and APC2, is sufficient to catalyze the ubiquitination of substrates securin and cyclin B1 (Tang *et al*, 2001). It has been shown that APC11 only is sufficient to ubiquitinate APC/C substrates under the presence of E1 and E2 ubiquitin ligase UBC4 (Gmachl *et al*, 2000; Leverson *et al*, 2000). Our study clearly claimed that APC11 but not APC2 is able to interact and degrade p65, thus control the osteogenesis of BMSCs through NF-κB pathway. Our future work would identify the E1 and E2 ubiquitin ligase responsible for APC11-triggered p65 ubiquitination.

The link between ubiquitination and the regulation of NF-κB pathway has been previously issued. Ubiquitin proteasome-dependent degradation of p65 is important for efficient and prompt termination of NF-κB activation (Hayden & Ghosh, 2004; Hou *et al*, 2010). Ubiquitination of p65 occurs majorly in the nucleus and inhibition of proteasome activity seems to selectively stabilize nuclear p65 protein (Saccani *et al*, 2004). Ubiquitination of p65 is regulated by different E3 ligases according to the context of cell type or signal. Many E3s have been found to promote the ubiquitination and degradation of p65, such as SOCS1 (suppressor of cytokine signaling 1) (Ryo *et al*, 2003), PPARγ (peroxisome proliferator-activated receptor-γ) (Hou *et al*, 2012), and COMMD1 (COMM domain containing 1) (Maine *et al*, 2007). Our present study demonstrated that APC11$^{CDC20}$ was a novel E3 ligase for ubiquitination and degradation of p65.

Sp7 (Osx) is one member of zinc-finger-containing transcription factor regulated by Runx2, which is required in osteoblast differentiation and bone formation (Nakashima *et al*, 2002; Celil & Campbell, 2005; Zhou *et al*, 2010). The Cre/loxP system has been used to generate conditional gene knockout mice commonly (Rodda & McMahon, 2006; Ono *et al*, 2014). In our study, breeding the *Sp7-Cre* mice with mice possessing Cdc20 flanked by loxP sites produces a mouse with Cdc20 gene deletion in Osx expressing cells and skeletal related tissues. *Sp7-Cre; Cdc20$^{f/f}$* mice represented reduced bone volume in long bone and reduction in new bone formation, illustrating the pivotal role of Cdc20 on bone formation *in vivo*. Additionally, our findings showed that CDC20 binds p65 and degrades it effectively through proteasome-ubiquitination pathway and may provide novel evidence for CDC20 functions as an inducer of osteogenic differentiation of BMSCs. Furthermore, the *in vitro* and in cells results demonstrated that CDC20 manipulated bone formation is NF-κB p65-dependent. This newly discovered function and mechanisms of CDC20 could help explain how E3s manipulate the osteogenesis differentiation of BMSCs and pave the way for developing therapeutic strategies of bone-related diseases.

# Materials and Methods

### Reagents and Tools table

| Reagent/Resource | Reference or Source | Identifier or Catalog Number |
|---|---|---|
| **Experimental Models** | **Source (public):** | |
| C57BL/6 (*M. musculus*) | Beijing Biocytogen Co., Ltd | C57BL/6-Sp7tm1(iCre)/Bcgen |
| hBMSCs | ScienCell Research Laboratories | Cat #7500 |
| **Recombinant DNA** | | |
| MT-CDC20 | Addgene | Cat #11593 |
| pcDNA3-HA-APC11 | Addgene | Cat #19898 |
| pBlueBac human APC2 | Addgene | Cat #11425 |
| pRSV-p65 | Addgene | Cat #20994 |
| **Antibodies** | | |
| Rabbit anti-α-tubulin | Proteintech | Cat #11224-1-AP |
| Rabbit anti-GAPDH | Proteintech | Cat #10494-1-AP |
| Rabbit anti-CDH1 | Proteintech | Cat #20874-1-AP |
| Rabbit anti-Ubiquitin | Proteintech | Cat #10201-2-AP |
| Rabbit anti-OCN | Proteintech | Cat #23418-1-AP |
| Rabbit IgG | Proteintech | Cat #B900610 |
| Mouse IgG | Proteintech | Cat #B900620 |

**Reagents and Tools table**   (continued)

| Reagent/Resource | Reference or Source | Identifier or Catalog Number |
|---|---|---|
| HRP-Mouse Anti-Rabbit IgG Light Chain Specific | Proteintech | Cat #SA00001-7L |
| Mouse anti-HA-Tag | Proteintech | Cat #66006-1-Ig |
| Mouse anti-His-Tag | Proteintech | Cat #66005-1-Ig |
| IPKine HRP AffiniPure Goat Anti-Mouse IgG Light Chain | Abbkine | Cat #A25012 |
| Rabbit anti-p65 | Abcam | Cat #ab16502 |
| Mouse anti-p65 | Cell Signaling Technology | Cat #6956T |
| Rabbit anti-Cyclin B1 | Abcam | Cat #ab181593 |
| Rabbit anti-CDC20 | Abcam | Cat #ab183479 |
| Rabbit anti-IκBα | Abcam | Cat #ab32518 |
| Goat anti-mouse | Abcam | Cat #ab6789 |
| Goat anti-rabbit | Abcam | Cat #ab190492 |
| Rabbit anti-RUNX2 | Cell Signaling Technology | Cat #12556 |
| Rabbit anti-H3 | Cell Signaling Technology | Cat #2650S |
| Rabbit anti-APC2 | Cell Signaling Technology | Cat #12301S |
| Rabbit anti-APC11 | Cell Signaling Technology | Cat #14090 |
| Mouse anti-Flag-tag | Cell Signaling Technology | Cat #8146 |
| Mouse anti-Myc-tag | Cell Signaling Technology | Cat #2276 |
| Rabbit anti-GST | Cell Signaling Technology | Cat #2622 |
| Rabbit anti-phospho-p65 | Cell Signaling Technology | Cat #3033 |
| Goat anti-mouse | ZSGB-BIO | Cat # ZF-0313 |
| Goat anti-rabbit | ZSGB-BIO | Cat # ZF-0311 |
| Rabbit anti-COL1A1 | Serviebio | Cat #GB11022-1 |
| Rabbit anti-COL2A1 | Serviebio | Cat #GB11021 |
| Rabbit anti-OPN | Serviebio | Cat #GB11500 |
| **Oligonucleotides and other sequence-based reagents** | | |
| PCR primers | This study | Table 2 |
| shRNA sequences | This study | Table 2 |
| siRNA sequences | This study | Table 2 |
| sgRNA sequences | This study | Table 1 |
| **Chemicals, Enzymes and other reagents** | | |
| Recombinat Human TNF-α Protein | R&D | Cat #210-TA-020 |
| MG132 | APE×BIO | Cat #A2585 |
| BAY 11-7082 | Selleck | Cat #S2913 |
| **Software** | | |
| GraphPad software | www.graphpad.com | |
| ImageJ | https://imagej.net/ImageJ | |
| **Other** | | |

**Methods and Protocols**

*Isolation and culture of cells*
Human bone marrow mesenchymal stromal/stem cells (hBMSCs) were obtained from ScienCell Research Laboratories (Carlsbad, CA, USA). hBMSCs from three different donors were used for cell-based *in vitro* studies. HEK293T cells were obtained from American Type Culture Collection (ATCC). Primary mouse BMSCs were separated from 6-week-old mice through flushing the bone marrow and collagenase digestion of tibiae and femurs. Mice were sacrificed and femurs and tibiae were collected. Small cuts (approximately 1–2 mm) were made at both proximal and distal ends of the bones and bone marrow was flushed into the collection tubes. Then, bones were cut up and digested in collagenase type 2 and type 4 (15 mg/ml, Worthington Biochemical Corporation) for 45 min. The supernatants of the digestion were mixed together with the flushed bone marrow for centrifugation, and the cell pellets were placed into the suitable dishes. After 48 h of culture, media was removed, aspirated

and non-adherent cells were discarded. The cells were digested and the cell pellets were resuspended in the volume of culture media for plating. When the BMSCs reached confluency, they were proceeded to perform differentiation (Maridas et al, 2018). The proliferation media (PM) comprises α-MEM (Invitrogen, Carlsbad, CA, USA) consisting of 10% fetal bovine serum (FBS) (PAA Laboratories GmbH, Linz, Austria), as well as 1% penicillin/streptomycin (Invitrogen). To induce osteogenic differentiation, cells were cultured in osteogenic media (OM) containing standard PM added with 0.2 mM ascorbic acid, 10 mM β-glycerophosphate, and 100 nM dexamethasone.

## Mice

Cdc20$^{f/f}$ and Sp7-Cre; Cdc20$^{f/f}$ mice were generated by Biocytogen Co., Ltd. (Beijing, China) using CRISPR/Cas9 based EGE system. Briefly, single guide RNAs (sgRNAs) targeting the upstream of exon1 and the downstream of exon7 of Cdc20, respectively, were designed using the CRISPR design tool (http://www.sanger.ac.uk/htgt/wge/). The candidate sgRNAs were screened for on-target activity using the UCA™ CRISPR efficiency evaluation kit (Biocytogen Co., Ltd., Beijing, China), and two sgRNAs with high specificity and on-target activity were selected. The targeting vector that contains the genomic DNA spanning exon1-7 of mouse Cdc20 flanked by two loxP sites and the homology arms at the 5′ and 3′ regions were constructed as well. The targeting vector, the in vitro-synthesized sgRNAs and Cas9 mRNA were co-injected into C57BL/6N mouse zygotes. After injection, the surviving 2 cell stage zygotes were transplanted into the KM albino pseudopregnant females. The founder mice bearing the floxed Cdc20 allele were determined by PCR amplification and DNA sequencing. Heterozygous Cdc20$^{fl/+}$ mice were obtained by crossing the founder mice and the wild-type C57BL/6N mice. The genotype of F1 heterozygous Cdc20$^{fl/+}$ mice was confirmed by PCR amplification, DNA sequencing, and Southern blot analysis. Sp7-Cre mice were generated from Beijing Biocytogen Co., Ltd. To avoid disrupting Sp7 expression, iCre will be inserted between the coding sequence of exon2 and 3′UTR. 2A will be used to achieve the expression of Sp7 and iCre at the same time and level. To avoid disrupting poly (A) signal of Sp7 and promoter region of the downstream gene Aaas, Neo cassette flanked by frt sites will be inserted within the non-conserved region of intron. Genotyping primers: Sp7-iCre-WT-F TACCAGAAGCGACCAC-TTGAGC; Sp7-iCre-Mut-R, GCACACAGACAGGAGCATCTTC. Primers were shown in Table 1. The detailed information of the iB-Sp7-iCre mice was shown on this website: https://biocytogen.com/products/cre-mouse-rat-models/b-sp7-osx-icre-mice/. This Sp7-Cre (Osx-Cre) mouse model has been used to examine the effects of specific gene loss in osteoblast progenitor cells on bone development (Jin et al, 2017; Chen et al, 2021). The application of this mouse model largely verifies that the Sp7-Cre transgene is able to specifically target Osx+ osteoprogenitors. The Institutional Animal Care and Use Committee of the Peking University Health Science Center (LA2014233) approved the conduction of this study; all the experiments were carried out through Institutional Animal Guidelines.

## Plasmid transfection and RNA interference

The plasmids MT-CDC20 (plasmid#11593), pcDNA3-HA-APC11 (plasmid#19898), pBlueBac human APC2 (plasmid#11425), and pRSV-p65 (plasmid#106453) were searched in ADDGENE website

### Table 1. sgRNA sequences and genotyping primers.

| Cdc20 sgRNA1 | GGTGGGATTACGTAAAATTAAGG | Validation |
|---|---|---|
| Cdc20 sgRNA2 | AAAGAAATTGGGTCCGAATGAGG | Validation |
| Cdc20 sgRNA3 | CCTAAACTATGTGGAGTTCAAGG | Validation |
| Cdc20 sgRNA4 | AGGTAGGGCTATGTAATAGAGGG | Validation |
| Cdc20 sgRNA5 | AACAGTTTACGTACATATTTTGG | Validation |
| Cdc20 sgRNA6 | CCTTGAACTCCACATAGTTTAGG | Validation |
| Cdc20 sgRNA7 | CAAGGTCTTCCTAAACTATGTGG | Validation |
| Cdc20 sgRNA8 | AGGTGACTCCCATTTAGTGCAGG | Validation and injection |
| Cdc20 sgRNA9 | CTGACATCTCCGTCTCGCCCTGG | Validation and injection |
| Cdc20 sgRNA10 | CAGGGCGAGACGGAGATGTCAGG | Validation |
| Cdc20 sgRNA11 | TTTTCAAACAACTCCAAGACAGG | Validation |
| Cdc20 sgRNA12 | GAGGAGCAGCCAGGGCGAGACGG | Validation |
| Cdc20 sgRNA13 | GACGGAGATGTCAGGTTACAGGG | Validation |
| Cdc20 sgRNA14 | AGACGGAGATGTCAGGTTACAGG | Validation |
| Cdc20 sgRNA15 | ATCTCCTGGAGACTCCCTCCTGG | Validation |
| Cdc20 sgRNA16 | GTCACCAGGAGGGAGTCTCCAGG | Validation |
| Genotyping | | |
| Cdc20 5′loxP-F | CCTAAACTATGTGGAGTTCAAGGCCA | WT:202 bp |
| Cdc20 5′loxP-R | AGGATCTAGGATCTAGGTGACTCCC | Mut:321 bp |
| Cdc20 3′loxP-F | GAAGCAGCTCCTGTCTTGGAGTTGT | WT:405 bp |
| Cdc20 3′loxP-R | CCACAGCCTGGGTGGAATGGATAAA | Mut:490 bp |
| Sp7-Cre WT-F | TACCAGAAGCGACCACTTGAGC | 263 bp |
| Sp7-Cre WT-R | CGCCAAGAGAGCCTGGCAAG | 263 bp |
| Sp7-Cre Mut-F | TACCAGAAGCGACCACTTGAGC | 445 bp |
| Sp7-Cre Mut-R | GCACACAGACAGGAGCATCTTC | 445 bp |

and purchased from BEIJING ZHONGYUAN LTD. All the plasmids were confirmed by DNA sequencing. For transient infection, cells at 70–80% confluency were transfected with plasmids through the Lipofectamine 3000 transfection kit (Invitrogen). Small-interfering RNAs (siRNA) targeting CDC20, APC2, APC11, p65, and the negative control (NC) were purchased from GenePharma Company (Shanghai, China). For transient infection, cells were transfected with siRNAs using Lipofectamine 3000 (Invitrogen) at 50–60% confluency according to the manufacturer's procedure, and after 36 h, cells were harvested for analyses. The oligonucleotides sequences are shown in Table 2.

## Lentiviral transfection

Recombinant lentivirus targeting CDC20, CDH1, APC11, APC2, p65, and the negative control (NC) were purchased from GenePharma Company (Shanghai, China). According to the manufacturer's guidance, BMSCs were infected with the viral supernatants at a MOI of 100 with the presence of polybrene (6 µg/ml). Puromycin (1 mg/ml; Sigma-Aldrich, USA) was used to select infected cells. The oligonucleotides sequences are shown in Table 2. For p65 lentivirus injections, $8 × 10^8$ TU/ml lentiviruses 100 µl every 2 weeks were injected for twice via the tail vein. Four weeks later, the mice were sacrificed.

**Table 2. Sequences of RNA and DNA oligonucleotides.**

| Name | Sense strand/ sense primer (5′–3′) | Antisense strand/ antisense primer (5′–3′) |
|---|---|---|
| Primers | | |
| *GAPDH* | AAGGAGTAAGACCCCTGGACCA | GCAACTGTGAGCAGGGGAGATT |
| *RUNX2* | ACTACCAGCCACCGAGACCA | ACTGCTTGCAGCCTTAAATGACTCT |
| *ALP* | TGTGTGGGGTGAAGGCCAAT | TCGTGGTGGTCACAATGCCC |
| *OCN* | CACCATGAGAGCCCTCACACTC | CCTGCTTGGACACAAAGGCTGC |
| *CDC20* | TGTGTGGCCTAGTGCTCCTG | ACACCATGCTACGGCCTTGA |
| *p65* | GCGCATCCAGACCAACAACAA | CCCACGCTGCTCTTCTTGGAA |
| *IL-6* | CGCAACAACTCATCTCATTCTGCG | CATGCTACATTTGCCGAAGAGC |
| *IL-8* | CGGATAAAGGGCCAAGAGAATATCCG | TCACATTCTAGCAAACCCATTCAA |
| *ICAM1* | ATTCAAGCTTAGCCTGGCCG | ATTTCCGGACTGACAGGGTG |
| *Gapdh* | CAGGAGAGTGTTTCCTCGTCC | TGAAGGGGTCGTTGATGGCA |
| *Cdc20* | CTCAAAGGACACACAGCACGG | CGCCACAACCGTAGAGTCTCA |
| *p65* | GCACACTGTGTTGCCCACTT | ACCCCTGGATGACGGTACAC |
| *Runx2* | GCAAGAAGGCTCTGGCGTTT | ACTGCTTGCAGCCTTAAATATTCCT |
| shRNA | | |
| Control | TTCTCCGAACGTGTCACGT | |
| CDC20-sh1 | GCACTGGACAACAGTGTGTAC | |
| CDC20-sh2 | GCAGAAACGGCTTCGAAATAT | |
| CDH1-sh1 | GCACTGGACAACAGTGTGTAC | |
| CDH1-sh2 | GGTTCAAGCTGCTGACCTTCA | |
| APC11sh | GCCAGGAATGGAAGTTCAAGG | |
| APC2sh | GCCGTGGGCATGCTGCGTAAT | |
| *p65*sh | CTCAAGATCTGCCGAGTAAAC | |
| siRNA | | |
| CDC20-si1 | GCAAAUCCAGUUCCAAGGUUU | ACCUUGGAACUGGAUUUGCUU |
| CDC20-si2 | CCUUGUGGAUUGGAGUUCUUU | AGAACUCCAAUCCACAAGGUU |
| *p65*si | GCGACAAGGUGCAGAAAGAUU | UCUUUCUGCACCUUGUCGCUU |
| p65si | GGAUUGAGGAGAAACGUAAUU | UUACGUUUCUCCUCAAUCCUU |
| APC11-si1 | GCAUCUGCAGGAUGGCAUUUU | AAUGCCAUCCUGCAGAUGCUU |
| APC11-si2 | CCUUGGUGCCUUGACCAUUUU | AAUGGUCAAGGCACCAAGGUU |
| APC2-si1 | GCUGCUGCAGGUUGACUAUUU | AUAGUCAACCUGCAGCAGCUU |
| APC2-si2 | CCAGCAUCCUUCGGAACUUUU | AAGUUCCGAAGGAUGCUGGUU |
| Negative control | UUCUCCGAACGUGUCACGUUU | ACGUGACACGUUCGGAGAAUU |

shRNA, short-hairpin RNA; siRNA, short interfering RNA.

### RNA isolation and quantitative RT–PCR

Total RNA was collected with TRIzol reagent (Invitrogen) and reverse-transcribed into cDNA with a PrimeScript RT Reagent Kit (Takara, Tokyo, Japan; #RR037A). Quantitative RT–PCR was conducted with SYBR Green Master Mix (Roche Applied Science, Mannheim, Germany), the 7500 Real-Time PCR Detection System (Applied Biosystems, Foster City, CA, USA) using GAPDH for normalization. The primers sequences are shown in Table 2.

### Western blot analyses

Cells were gathered and lysed in RIPA buffer (1 mM EDTA, 10 mM Tris–HCl, 1% NP-40, 1% sodium dodecyl sulfate (SDS), 1:100 PMSF, 1:100 proteinase inhibitor cocktail, 50 mM β-glycerophosphate,

50 mM sodium fluoride). Proteins were separated on a 15, 10, or 7.5% SDS–polyacrylamide gel and transferred to polyvinylidene difluoride (PVDF) membranes. After membranes were blotted with 5% dehydrated milk for 1–3 h, the primary antibodies against RUNX2 (Cell Signaling Technology, 12556), Cyclin B1 (Abcam, ab181593), p65 (Abcam, ab16502), α-tubulin (Proteintech, 11224-1-AP), CDH1 (Proteintech, 20874-1-AP), GAPDH (Proteintech, 10494-1-AP), Ubiquitin (Proteintech, 10201-2-AP), Rabbit IgG (Proteintech, B900610), Mouse IgG (Proteintech, B900620), HA-Tag (Proteintech, 66006-1-Ig), His-Tag (Proteintech, 66005-1-Ig), H3 (Cell Signaling Technology, 2650S), CDC20 (Abcam, ab183479), APC2 (Cell Signaling Technology, 12301S), APC11 (Cell Signaling Technology, 14090), Myc-tag (Cell Signaling Technology, 2276), Flag-tag (Cell

Signaling Technology, 8146), GST (Cell Signaling Technology, 2622), Phospho-p65 (Cell Signaling Technology, 3033), and IκBα (Abcam, ab32518) were diluted according to the instructions and incubated with these membranes at 4°C overnight. Then, anti-mouse (Abcam, ab6789), anti-rabbit (Abcam, ab190492), anti-rabbit IgG Light Chain Specific (Proteintech, SA00001-7L), and anti-mouse IgG Light Chain (Abbkine, A25012) secondary antibodies were diluted 1:10,000 and incubated with the membranes at room temperature for 1–3 h. The membranes were visualized through an ECL kit (CWBIO), and the band intensities were quantified using ImageJ software (NIH, Bethesda, MD, USA). The signal of the target band was normalized to that of GAPDH, α-tubulin, or H3 band.

### Extraction of cytoplasmic and nuclear protein

Cells were collected and suspended in buffer A (0.1 mM EDTA, 0.1 mM EGTA, 10 mM HEPES, 1 mM DTT, 10 mM KCl, 0.15% NP-40, and 1:100 proteinase inhibitor cocktail) on ice for 10 min. The supernatants of the samples were collected as the cytoplasmic protein after centrifugation. The remained pellets were washed using phosphate-buffered saline (PBS) and suspended in buffer B (1 mM EGTA, 1 mM EDTA, 1 mM DTT, 20 mM HEPES, 400 mM NaCl, 0.5% NP-40, and 1:100 proteinase inhibitor cocktail) at 4°C for 25 min. The supernatants were gathered and used as the nuclear protein after centrifugation.

### Immunoprecipitation assays

Cells were collected and lysed in a Nonidet P-40 buffer together with a complete protease inhibitor mixture (Roche). The lysates were used for immunoprecipitation with the indicated antibodies. Generally, 4–6 µg of antibodies were incubated with approximately 800 mg of cell lysates at 4°C for 8–12 h. After that, Protein A/G-agarose beads were added in the lysates and incubated at 4°C for 2–4 h. Then, the immunoprecipitations were washed in lysis buffer and eluted with 2× SDS loading buffer through boiling for 5 min at 99°C. All the data were collected from at least three independent experiments.

### In vivo and in vitro ubiquitination assay

For *in vivo* ubiquitination assays, HEK293T cells were co-transfected with different combination of Flag-p65, Myc-CDC20, HA-APC11, HA-Ubiquitin, His-Ubiquitin, and Vector plasmids to assess the *in vivo* ubiquitination of p65. Cells were harvested after 36 h of transfection with the addition of 10 µM MG132 treatment for 6 h. Cells were collected and lysed; 800 µg of whole-cell lysates were immunoprecipitated using anti-Flag antibody. The resultant immunoprecipitants were resolved in SDS–PAGE and probed for anti-HA or anti-His antibodies to assess the relative ubiquitinated levels of p65. Endogenous ubiquitination was probed for ubiquitin antibody in NC, CDC20sh, APC11sh, APC2sh, CDH1sh, *Cdc20^{f/f}*, and *Sp7-Cre; Cdc20^{f/f}* cell lysates. The *in vitro* ubiquitination assays were conducted in 20 µl ubiquitination assay buffer [50 mM Tris–HCl (pH 7.4), 50 mM NaCl, 50 mM MgCl₂, 1.5 mM dithiothreitol, and 10 mM ATP], with 0.7 µg E1, 1 µg UbcH10 (E2), 15 µg HA-Ubiquitin (all from Boston Biochem), 0.7 µg Flag-p65, and 1.5 µg GST or GST-CDC20 or GST-APC11, and the reaction mixture was incubated at 32°C for 1 h. Supernatant was gathered from insoluble material by centrifugation. The reaction was terminated by adding SDS sample buffer. The reaction products were resolved by SDS–PAGE and probed with the indicated antibodies.

### GST pull-down assay

CDC20 and p65 cDNA were cloned into PGEX-6P-1 vector and were identified by DNA sequencing. GST-CDC20 and GST-p65 were expressed in E. coli strain BL21 (DE3). Equal amounts of GST or GST fusion proteins were incubated with 50% glutathione sepharose (Sangon Biotech) at 4°C for 4 h. Then, the mixture was washed in buffer [20 mM Tris (pH 7.4), 0.1 mM EDTA, 100 mM NaCl] three times. The resulting GST-p65 or GST-CDC20 were incubated with cell extracts separately. The beads were washed with buffer [0.1 mM EDTA, 0.5% Nonidet P-40, 20 mM Tris (pH 7.4) and 300 mM NaCl] for three times. The bounded proteins were eluted by boiling in sample buffer and then analyzed by immunoblotting with anti-CDC20 or anti-p65 antibodies.

### Immunochemistry

For immunofluorescence in cells, hBMSCs were seeded in 24-well plates, cells grown on glass coverslips were fixed in 4% paraformaldehyde for 20 min and permeabilized using 0.25% Triton X-100 for 15 min. Then, cells were blocked with 0.8% BSA for 1 h. Next, cells were incubated with the primary antibodies (1:200) against CDC20 (Abcam, ab183479) and p65 (Cell Signaling Technology, 6956T) at 4°C overnight and incubated them in 1:500 specified secondary antibody (ZSGB-BIO, ZF-0313 and ZF-0311) for 1 h. Nuclei were stained with DAPI, and the coverslips were on a glass slide. Fluorescence staining was visualized with a Confocal Zeiss Axiovert 650 microscope. Images were shot with a LSM 5 EXCITER confocal imaging system (Carl Zeiss, Oberkochen, Germany). For immunofluorescence in sections, using similar methods, sections were examined by fluorescence microscopy. For 5-ethynyl-2′-deoxyuridine (EdU) immunofluorescence labeling, mice were sacrificed 3 h after intraperitoneal injection with EdU (5 mg/kg; RiboBio), and the incorporated EdU was detectable by the Click-iT Apollo 567 Stain Kit (RiboBio). For immunohistochemistry (IHC), paraffin sections were dewaxed and rehydrated using standard protocols. Sections were incubated with antibodies specific to OCN (Proteintech, 23418-1-AP), COL1A1 (Servicebio, GB11022-1), OPN (Servicebio, GB11500) overnight at 4°C, followed by washing and incubating with secondary antibodies. Nuclei were counterstained with DAPI.

### ALP staining and quantification

BMSCs were cultured in six-well plates for 7 days in proliferation medium (PM) or osteogenic medium (OM). Cells were washed with PBS and then fixed in 4% paraformaldehyde at room temperature (RT) for 15 min and washed with PBS three times. Subsequently, cells were incubated with a 5-bromo-4-chloro-3-indolyl phosphate-4-nitro blue tetrazolium (BCIP/NBT) staining kit (CWBIO, Beijing, China) solution for 15 min at room temperature and then rinsed with distilled water. For quantification of ALP activity, it was assayed with an ALP activity kit using 5 µl protein lysates according to the manufacturer's procedure (Sigma-Aldrich, USA). Signals were normalized based on protein content.

### Bone formation analyses

The BMSCs transfected with lentivirus (NC, CDC20sh-1, CDC20sh-2) were resuspended with 40 mg Synthograft™, a beta-tricalcium phosphate (β-TCP) (β-tricalcium phosphate; Bicon) for 1 h at 37°C, followed by centrifugation at 150 × *g* for 5 min. The scaffolds were

implanted in two separate symmetrical sites on the dorsal subcutaneous sites of 6-week-old BALB/c homozygous nude (nu/nu) mice. Eight weeks later, implants were collected and fixed in 4% paraformaldehyde and then decalcified in 10% EDTA (pH = 7.4), followed by dehydration and embedded with paraffin. The five-μm-thick sections were cut and stained with H&E and Masson's trichrome together with IHC assay. All animal experiments were approved by the Institutional Animal Care and Use Committee of the Peking University Health Science Center (LA2014233).

### Micro-CT analyses of mice

The collected bone tissues were fixed in 4% paraformaldehyde for 48 h and stored in 75% ethanol. The specimens were scanned using an Inveon MM system (Siemens, Munich, Germany). Images were obtained at voltage of 80 kV, current of 500 μA, effective pixel size of 8.82 μm, and exposure time of 1,500 ms of every 360 rotational steps. For quantification and analyses of the images, bone mineral density (BMD), bone volume/total volume (BV/TV), trabecular number (Tb.N), and trabecular spacing (Tb. Sp) parameters were analyzed using Image-Pro Plus software (Media Cybernetics, Rockville, MD).

### Histology and staining

The femurs and tibiae of mice were isolated for histology, fixed in phosphate-buffered formalin for 2 days followed by decalcification in 0.5 M ethylenediaminetetraacetic acid (EDTA) pH 7.8 for 3 weeks, and then embedded in paraffin using standard procedures. Before histochemistry, prepared 5 μm tissue sections on slides were deparaffinized and rehydrated using standard protocols. Sections for distal femurs and growth plates were conducted with Goldner's trichrome staining. The cartilages and endochondral bones were stained with Safranin-O (0.05%) and counterstained with Fast Green (0.1%). The liver and distal femurs were stained with hematoxylin and eosin (H&E). The whole skeletal of postnatal mice were stained with Alcian Blue followed by Alizarin red. For determination of osteoclasts, sections were stained with tartrate-resistant acid phosphatase (TRAP) (pH 4.7–5.0 at 37°C). For the analysis of osteoblastic and osteoclastic activity, serum concentrations of P1NP and CTX-1 were measured using ELISA kits (Jiangsu Meimian industrial Co., Ltd). Detection of apoptosis was performed by a TUNEL kit (Servicebio, G1507-50) according to the manufacturer's instructions. ImageJ software (NIH, Bethesda, MD, USA) was used to quantify intensity and cell numbers.

### Tibial cortical defect model

For tibial cortical bone defect, the 0.8 mm diameter holes were produced on the anterior surface of the tibiae with a round bur (Komet, Germany) conducting at 1,000 rpm with saline irrigation at about 2 mm under the knees. These defects were conducted on 12-week-old mice, and these mice were sacrificed 10 days later.

### Statistical analyses

All statistical calculations were conducted through the GraphPad software (GraphPad Software, Inc., La Jolla, CA, USA). All the data were presented as the mean ± standard deviation (SD) of at least three experiments per group. Differences between two groups were carried out by independent two-tailed Student's $t$-tests, and comparisons between more than two groups were calculated by one-way ANOVA followed by a Tukey's post hoc test. A two-tailed value of $P < 0.05$ was considered statistically significant.

## Data availability

This study includes no data deposited in external repositories.

**Expanded View** for this article is available online.

## Acknowledgements

The project was supported by the Beijing Natural Science Foundation (7182183 to YZ. and 7202233 to P.Z.) and the National Natural Science Foundation of China (87930026 to YZ. and 81970911 to P.Z.).

## Author contributions

PZ, YZ: design and conception, manuscript writing, financial support, final approval of manuscript; YD: laboratory work, data collection and analysis, manuscript writing; MZ, XL, ZL, MH, YT, XZ, LL, YL, laboratory work, assembly of data, provision of study material and final approval of manuscript.

## Conflict of interest

The authors declare that they have no conflict of interest.

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
