## [Review Process File · EMBO Reports]

CDC20 promotes bone formation via APC/C dependent ubiquitination and degradation of p65

Yangge Du, Min Zhang, Xuejiao Liu, Zheng Li, Menglong Hu, Yueming Tian, Longwei Lv, Xiao Zhang, Yunsong Liu, Ping Zhang and Yongsheng Zhou

DOI: 10.15252/embr.202152576

Corresponding author(s): Yongsheng Zhou (kqzhouysh@hsc.pku.edu.cn) , Ping Zhang (zhangping332@bjmu.edu.cn)

Review Timeline:

Submission Date:	31st Jan 21
Editorial Decision:	3rd Mar 21
Revision Received:	26th May 21
Editorial Decision:	16th Jun 21
Revision Received:	23rd Jun 21
Accepted:	8th Jul 21

Editor: Deniz Senyilmaz Tiebe

Transaction Report:

Dear Dr. Zhou,

Thank you for the submission of your research manuscript to our journal, which was now seen by three referees, whose reports are copied below.

Referees express interest in the proposed role of CD20 in osteogenesis by promoting APC/C mediated p65 degradation. However, they also raise overlapping important concerns that need to be addressed to consider publication here.

I find the reports informed and constructive, and believe that addressing the concerns raised will significantly strengthen the manuscript. As the reports are below, and I think all points need to be addressed, I will not detail them here.

Given these positive recommendations, we would like to invite you to revise your manuscript with the understanding that the referee concerns (as in their reports) must be fully addressed and their suggestions taken on board. Please address all referee concerns in a complete point-by-point response. Acceptance of the manuscript will depend on a positive outcome of a second round of review. It is EMBO reports policy to allow a single round of revision only and acceptance or rejection of the manuscript will therefore depend on the completeness of your responses included in the next, final version of the manuscript.

*** Temporary update to EMBO Press scooping protection policy:

We are aware that many laboratories cannot function at full efficiency during the current COVID-19/SARS-CoV-2 pandemic and have therefore extended our 'scooping protection policy' to cover the period required for a full revision to address the experimental issues highlighted in the editorial decision letter. Please contact the scientific editor handling your manuscript to discuss a revision plan should you need additional time, and also if you see a paper with related content published elsewhere.***

1. A data availability section providing access to data deposited in public databases is missing (where applicable).
2. Your manuscript contains statistics and error bars based on $n=2$. Please use scatter plots in these cases.

You can submit the revision either as a Scientific Report or as a Research Article. For Scientific Reports, the revised manuscript can contain up to 5 main figures and 5 Expanded View figures. If the revision leads to a manuscript with more than 5 main figures it will be published as a Research Article. In this case the Results and Discussion section should be separate. If a Scientific Report is submitted, these sections have to be combined. This will help to shorten the manuscript text by

eliminating some redundancy that is inevitable when discussing the same experiments twice. In either case, all materials and methods should be included in the main manuscript file

Supplementary/additional data: The Expanded View format, which will be displayed in the main HTML of the paper in a collapsible format, has replaced the Supplementary information. You can submit up to 5 images as Expanded View. Please follow the nomenclature Figure EV1, Figure EV2 etc. The figure legend for these should be included in the main manuscript document file in a section called Expanded View Figure Legends after the main Figure Legends section. Additional Supplementary material should be supplied as a single pdf labeled Appendix. The Appendix includes a table of content on the first page with page numbers, all figures and their legends. Please follow the nomenclature Appendix Figure Sx throughout the text and also label the figures according to this nomenclature. For more details please refer to our guide to authors.

Please note that for all articles published beginning 1 July 2020, the EMBO Reports reference style will change to the Harvard style for all article types. Details and examples are provided at <https://www.embopress.org/page/journal/14693178/authorguide#referencesformat>

2) individual production quality figure files as .eps, .tif, .jpg (one file per figure).

3) a .docx formatted letter INCLUDING the reviewers' reports and your detailed point-by-point responses to their comments. As part of the EMBO Press transparent editorial process, the point-by-point response is part of the Review Process File (RPF), which will be published alongside your paper. For more details on our Transparent Editorial Process, please visit our website: <https://www.embopress.org/page/journal/14693178/authorguide#transparentprocess>
You are able to opt out of this by letting the editorial office know (emboreports@embo.org). If you do opt out, the Review Process File link will point to the following statement: "No Review Process File is available with this article, as the authors have chosen not to make the review process public in this case."

4) a complete author checklist, which you can download from our author guidelines (<http://embor.embopress.org/authorguide>). Please insert information in the checklist that is also reflected in the manuscript. The completed author checklist will also be part of the RPF.

5) Please note that all corresponding authors are required to supply an ORCID ID for their name upon submission of a revised manuscript (<https://orcid.org/>). Please find instructions on how to link your ORCID ID to your account in our manuscript tracking system in our Author guidelines (<http://embor.embopress.org/authorguide>).

6) We replaced Supplementary Information with Expanded View (EV) Figures and Tables that are collapsible/expandable online. A maximum of 5 EV Figures can be typeset. EV Figures should be cited as 'Figure EV1, Figure EV2' etc... in the text and their respective legends should be included in the main text after the legends of regular figures.

- For the figures that you do NOT wish to display as Expanded View figures, they should be

bundled together with their legends in a single PDF file called *Appendix*, which should start with a short Table of Content. Appendix figures should be referred to in the main text as: "Appendix Figure S1, Appendix Figure S2" etc. See detailed instructions regarding expanded view here: <<http://embor.embopress.org/authorguide#expandedview>>.

7) We would also encourage you to include the source data for figure panels that show essential data.

Numerical data should be provided as individual .xls or .csv files (including a tab describing the data). For blots or microscopy, uncropped images should be submitted (using a zip archive if multiple images need to be supplied for one panel). Additional information on source data and instruction on how to label the files are available <<http://embor.embopress.org/authorguide#sourcedata>>.

8) Our journal encourages inclusion of *data citations in the reference list* to directly cite datasets that were re-used and obtained from public databases. Data citations in the article text are distinct from normal bibliographical citations and should directly link to the database records from which the data can be accessed. In the main text, data citations are formatted as follows: "Data ref: Smith et al, 2001" or "Data ref: NCBI Sequence Read Archive PRJNA342805, 2017". In the Reference list, data citations must be labeled with "[DATASET]". A data reference must provide the database name, accession number/identifiers and a resolvable link to the landing page from which the data can be accessed at the end of the reference. Further instructions are available at <<http://embor.embopress.org/authorguide#datacitation>>.

9) Please make sure to include a Data Availability Section before submitting your revision - if it is not applicable, make a statement that no data were deposited in a public database. Primary datasets (and computer code, where appropriate) produced in this study need to be deposited in an appropriate public database (see <<http://embor.embopress.org/authorguide#dataavailability>>).

The accession numbers and database should be listed in a formal "Data Availability " section (placed after Materials & Method) that follows the model below. Please note that the Data Availability Section is restricted to new primary data that are part of this study.

Data availability

10) Regarding data quantification, please ensure to specify the name of the statistical test used to generate error bars and P values, the number (n) of independent experiments underlying each data point (not replicate measures of one sample), and the test used to calculate p-values in each figure legend. Discussion of statistical methodology can be reported in the materials and methods section, but figure legends should contain a basic description of n, P and the test applied.

Please note that error bars and statistical comparisons may only be applied to data obtained from at least three independent biological replicates.

I look forward to seeing a revised version of your manuscript when it is ready. Please let me know if you have questions or comments regarding the revision.

Yours sincerely,

Deniz Senyilmaz Tiebe

Deniz Senyilmaz Tiebe, PhD
Editor
EMBO Reports

Referee #1:

Yangge Du and collaborators present in vivo and in vitro data demonstrating a novel and important role of CDC20 in bone formation. Genetic ablation of Cdc20 in Osterix-expressing cells reduces bone mass of 6 and 12-week-old conditional knockout (CKO) mice. In vitro, bone marrow mesenchymal stromal cells (BMSCs) lacking CDC20 present reduced expression of the osteogenic transcription factor RUNX2 and reduced capacity to generate alkaline phosphatase (ALP)-expressing osteoblasts. Mechanistically, the authors show that CDC20 forms a complex with APC11 that ubiquitinates and targets p65 NFkB for proteasomal degradation. Their results obtained after pharmacological inhibition of the NFkB pathway (using BAY11-7082) and p65 knockdown (using shRNAs) in CDC20-deficient BMDCs, as well as p65 knockdown in CDC20 knockout mice (using systemic injections of shRNAs against p65) indicate that CDC20 regulates bone formation, at least in part through proteasomal degradation of p65.

This manuscript is of very good quality, with a lot of work overall well executed. However, there are a few important points that must be addressed, in particular to validate the mouse model used in this study. These points are listed below (see "major comments"). Additional minor points should also be considered to further improve the quality of the manuscript (see "minor comments" below).

Major comments:

1- There are important missing verifications and information concerning the materials and methods, notably concerning the mouse model used in the current study. The Sp7-Cre line differs from the

Osx-GFP::Cre (Tet-off) line that is classically used in the bone field. It seems that Sp7-Cre mice have not been previously characterized. No reference and no specific information are provided regarding this novel Cre line. In absence of complete description of the promoter used to drive Cre expression, and proper characterization of this mouse line, we do not know whether the Sp7-cre transgene targets specifically Osx⁺ osteoprogenitors (without off-sites effects), and how this line compares to Osx-GFP::Cre (tet-off) animals.

2- Tail vein/systemic injections of lentivirus encoding shRNAs to knockdown gene expression in mice are not commonly performed. Although we understand that lentiviruses were injected during 4 weeks in CDC20 CKO mice, we do not know how much virus was inoculated, and how often. This information must be written in the manuscript. Since lentiviruses are inoculated in a systemic fashion, various "side effects" of p65 knockdown could be induced, and could complicate the interpretation of the rescue experiment. Ideally p65 could be conditionally knocked-out of osteoprogenitors, since floxed p65 mice exist. Alternatively, the authors should provide evidence that mice inoculated with lentiviruses during 4 weeks are healthy, and that apoptosis of liver hepatocytes is not induced in these mice, since whole-body deficiency of p65 leads to massive liver apoptosis and death. p65 expression in the liver of experimental and control animals should be evaluated, and perhaps in other organs as well.

3- Additional controls and time-points should be presented in the manuscript in order to better characterize the skeletal phenotype of the Cdc20 CKO mice. The bone phenotype of Sp7-Cre; +/- (WT) and Sp7-Cre; fl/+ (conditional HETs) must be shown in addition to Cre-negative animals. Since Osterix is also expressed in growth plate chondrocytes, it is important to characterize the growth plate phenotype of the Sp7-Cre; fl/fl (CKO) mice. Part of the bone phenotype shown in 6-week-old mice could result from delayed endochondral bone formation, due to altered cartilage formation. It is important to know if this is the case. Characterization of endochondral bone development in embryos and newborn mice should provide valuable information. It would be interesting and important to provide data on the relative expression of CDC20 and p65 (mRNA and/or protein) in chondrocytes, osteoprogenitors, osteoblasts and osteocytes in wild type and CKO mice, on bone sections.

Minor comments:

1- Histological analyses of the scaffold engrafted subcutaneously in nude mice (Fig. 1J, and 1K) should be further analyzed by in situ hybridization with osteoblast markers (such as type I collagen and/or osteopontin), to support the conclusion that CDC20 knockdown in hBMDCs impairs osteoblast differentiation.

2- Fig 2E, indicates that the holes in tibias of control and CKO have been drilled on opposite sides of the bone. Since the cortical thickness varies depending on the side of the tibia, it is important to generate bone defects at the exact same location for all control and CKO mice. It is also important to specify at which age these defects have been created.

3- Fig. 6F, 3rd lane: there is a large APC11 band, while the legend indicates no HA-APC11 overexpressed in this sample. The legend appears correct since p65 protein is upregulated in this sample. The APC11 blot should show a band in the second lane, not the third lane. This implies that immunostaining of p65 and of APC11 has been done on two unrelated blots. The entire experiment should be done again using the same membrane to detect p65, APC11, CDC20 and GAPDH.

4- Fig. 7C shows robust induction of p65 ubiquitination in vitro in presence of purified CDC20 without APC11. Does this mean that APC11 is not required for CDC20-induced ubiquitination of p65?

5- Fig 7F, middle lane shows a strong downregulation of CDC20 protein expression although no shRNAs against CDC20 are expressed. How do you explain this?

6- The presentation of the results could be slightly reorganized by presenting first the mouse model

and its skeletal phenotype, and then in vitro experiments with BMDCs (currently presented in Fig. 1). Moreover, the presentation of interaction of CDC20 and p65 both in the text and in the figures appears somewhat lengthy and could be shortened. Finally, the results presented in fig. 6I-L could be transferred to figure 8, since they support the conclusion that p65/NFkB signaling plays a role downstream of CDC20 in osteoblast differentiation.

7- The protocol used to generate BMSCs is unclear.

Very minor additional comments:

- 1- The term "in vivo" page 5, line 108 is not appropriate since it refers to in vitro experiments.
- 2- Page 6, line 121 the text "decreased ALP staining and activity" could be replaced by "decreased ALP staining and quantification" since the staining actually reflects ALP activity.
- 3- The term "in vivo" page 13 line 282 may be misleading, since this is not observed in live mice but in cells cultured in vitro. This could be replaced by "in cells".

Referee #2:

The authors describe a novel function of CDC20, a co-activator of the E3 ubiquitin ligase APC/C, in bone remodeling and regeneration. Whereas in vivo data to demonstrate this function were obtained by the generation and analysis of mice with Sp7-Cre-mediated CDC20 inactivation in bone-forming osteoblasts, the majority of experiments were set up to address the molecular bases explaining the deduced function of CDC20 in bone formation. Overall, this is a very complex study with strong and potentially relevant data. There are however several issues that remain to be addressed.

Specific comments:

- 1) Although the reported results are surely relevant from a basic science perspective, the statement that APC11CDC20 is a novel target for osteoporosis treatment appears too strong. Is it truly realistic to establish an agonist of such a complex cytoplasmatic regulator, whose expression is not restricted to osteoblasts?
- 2) The authors have to provide more information about their Sp7-Cre model, which apparently has been generated for this specific project. The key question is, if these mice represent a Cre knockin into the Sp7 locus, which might interfere with the endogenous expression of the relevant transcription factor Osterix. If this is the case, the Cdc20^{f/f} mice alone would not be sufficient as a control, and it has to be evaluated if the Sp7-Cre mice display a skeletal phenotype, regardless of their Cdc20 genotype.
- 3) Since the in vivo findings in Sp7-Cre/Cdc20^{f/f} mice are essentially the basis for the molecular experiments described thereafter, it would be important to strengthen them. In fact, there was no analysis performed to understand the cellular causes of the osteopenia. This could be done by cellular histomorphometry quantifying osteoblast and osteoclast numbers on the femur sections. Alternatively, the authors could measure serum levels of bone formation and resorption biomarkers. As presented, the phenotype description is incomplete.
- 4) The molecular experiments shown in the second part of the manuscript are overall convincing, although I personally consider the data obtained in BMCSs (now mostly shown as Supplemental Figures) as more relevant than the data obtained in HEK cells. Moreover, although some of the Western blot results are confirmed by quantitative data, many others are not. Since the authors state that all data were collected from at least three independent experiments, it should be

possible to perform quantification and statistical analysis for all presented data.

5) In the last paragraph of the Results section the authors state that "..., lentiviruses were distributed in mice and the expression levels of p65 was knocked down according to qRT-PCR and western blot analyses." While this statement suggests that p65 expression was analyzed *in vivo*, the legend of Figure EV7B/C indicates that these results were obtained in BMSCs. If this means that the expression analysis was performed in *ex vivo* cultures, this has to be clearly stated.

6) Given the complexity of their findings, the authors should think about providing a schematic presentation to illustrate their conclusions.

Referee #3:

In this manuscript, Du et al. report a positive regulation of osteogenesis by the APC/C(Cdc20) ubiquitin E3 ligase complex through modulating p65/RelA stability. The authors observed a positive correlation between Cdc20 expression and osteogenesis using both *in vitro* BMSC culture/differentiation models and an *in vivo* conditional Cdc20 knockout mouse model. There is a decreased bone mass found in Cdc20-depleted animals. The authors then sought to define the underlying mechanism, they found p65 was upregulated in Cdc20-deficient BMSCs, hence hypothesized p65 as an APC(Cdc20) ubiquitin substrate. They provided evidence that p65 bound to Cdc20 and that Cdc20 facilitates p65 ubiquitination. To assess the role of p65 in Cdc20-mediated osteogenesis, the authors delivered shp65 lentivirus into the mice and were able to observe a rescued phenotype. This is an interesting study demonstrating a novel role of Cdc20 in osteogenesis. However, the data in their current form are quite preliminary, hence a major revision is recommended to provide strong evidence to support the conclusions. Below are specific comments that should be addressed prior to being considered for publication in EMBO Reports.

1. Cdc20 is essential for M->G1 transition, complete loss of Cdc20 arrests cell cycle in M phase and eventually causes cell death. Can the authors comment on whether the phenotypes observed are at least in part due to decreased osteoblast numbers upon *Osx/Sp7*-driven deletion of Cdc20?
2. Fig. 1H-I, can the authors comment on why the Cdc20 WD40 domain was able to rescue the decreased ALP activity upon Cdc20 loss?
3. Fig. 2, Did the authors perform whole-mount skeleton staining of control and *Sp7-cre;Cdc20(f/f)* mice?
4. Fig. 2, Have the authors tried co-staining Cdc20, p65 and an osteoblast marker using immunohistochemistry?
5. Fig. 3, Did the authors check p-p65 levels in conditions where Cdc20 is depleted?
6. Fig. 3, In siCdc20 cells, was p65 mRNA affected as well?
7. Fig. 3 and 4, Why did the authors choose to focus on the p65/NF- κ B pathway in this study? Are other NF- κ B pathway proteins found increased upon Cdc20 deletion or interacted with Cdc20?
8. Fig. 6, APC2 is required for the integrity of APC/C complex, did the authors suggest in this study an APC2-absent but APC11-present APC/C subcomplex to catalyze p65 ubiquitination? The authors should examine known APC(Cdc20) substrates to demonstrate an effective depletion of APC2, APC11, and Cdh1.
9. Fig. EV5, was Cdc20 upregulated when knocking down Cdh1?
10. Fig. 7C, bacterially expressed and purified GST-Cdc20 and GST-APC11 are APC/C-free, the authors need to explain why APC-free GST-Cdc20 and GST-APC11 promote the ubiquitination of p65 *in vitro*.

**Response to Editor and Reviewer**

Ms. Ref. No.: EMBOR-2021-52576

Title: CDC20 regulates bone formation via APC/C dependent ubiquitination and
degradation of p65

Dear Prof. Deniz Senyilmaz Tiebe,

We would like to express our sincere thanks to the editor and reviewers for the critical
comments and insightful suggestions concerning our manuscript. We found the
reviewers' comments to be very helpful and of great value for improving the quality
of our manuscript and we have now made a thorough revision of the paper based on
new experimental evidence. All the changes are highlighted in manuscript and a
detailed point-to-point response to the editor and reviewers were provided. We hope
that our revised manuscript and supporting information will meet the high standard of
*EMBO reports*.

**[Responses]**

Referee #1:

Yangge Du and collaborators present in vivo and in vitro data demonstrating a novel
and important role of CDC20 in bone formation. Genetic ablation of Cdc20 in
Osterix-expressing cells reduces bone mass of 6 and 12-week-old conditional
knockout (CKO) mice. In vitro, bone marrow mesenchymal stromal cells (BMSCs)
lacking CDC20 present reduced expression of the osteogenic transcription factor
RUNX2 and reduced capacity to generate alkaline phosphatase (ALP)-expressing

osteoblasts. Mechanistically, the authors show that CDC20 forms a complex with
APC11 that ubiquitinates and targets p65 NFkB for proteasomal degradation. Their
results obtained after pharmacological inhibition of the NFkB pathway (using
BAY11-7082) and p65 knockdown (using shRNAs) in CDC20-deficient BMDCs, as
well as p65 knockdown in CDC20 knockout mice (using systemic injections of
shRNAs against p65) indicate that CDC20 regulates bone formation, at least in part
through proteasomal degradation of p65.

This manuscript is of very good quality, with a lot of work overall well executed.
However, there are a few important points that must be addressed, in particular to
validate the mouse model used in this study. These points are listed below (see "major
comments"). Additional minor points should also be considered to further improve the
quality of the manuscript (see "minor comments" below).

Major comments:

1- There are important missing verifications and information concerning the materials
and methods, notably concerning the mouse model used in the current study. The
Sp7-Cre line differs from the Osx-GFP::Cre (Tet-off) line that is classically used in
the bone field. It seems that Sp7-Cre mice have not been previously characterized. No
reference and no specific information are provided regarding this novel Cre line. In
absence of complete description of the promoter used to drive Cre expression, and
proper characterization of this mouse line, we do not know whether the Sp7-cre

transgene targets specifically *Osx*+ osteoprogenitors (without off-sites effects), and
how this line compares to *Osx*-GFP::Cre (tet-off) animals.

**Response:** Thanks for the constructive points. Osterix (*Osx*, *Sp7*) is a specific
transcription factor of osteoprogenitors and we generated *Sp7-Cre;Cdc20^{fl/fl}* mice to
examine the role of CDC20 during osteoprogenitors differentiation and bone
formation. We have added the description about the mouse model in the methods and
materials in manuscript in page 23, line 493 as follows.

“*Cdc20^{fl/fl}* and *Sp7-Cre;Cdc20^{fl/fl}* mice were generated by Biocytogen Co., Ltd. (Beijing,
China) using CRISPR/Cas9 based EGE system. Briefly, single guide RNAs (sgRNAs)
targeting the upstream of exon1 and the downstream of exon7 of *Cdc20*, respectively,
were designed using the CRISPR design tool (<http://www.sanger.ac.uk/htgt/wge/>).
The candidate sgRNAs were screened for on-target activity using the UCA™
CRISPR efficiency evaluation kit (Biocytogen Co., Ltd., Beijing, China), and two
sgRNAs with high specificity and on-target activity were selected. The targeting
vector that contains the genomic DNA spanning exon1-7 of mouse *Cdc20* flanked by
two loxP sites and the homology arms at the 5' and 3' regions were constructed as
well. The targeting vector, the in vitro-synthesized sgRNAs and Cas9 mRNA were
co-injected into C57BL/6N mouse zygotes. After injection, the surviving 2-cell-stage
zygotes were transplanted into the KM albino pseudopregnant females. The founder
mice bearing the floxed *Cdc20* allele were determined by PCR amplification and
DNA sequencing. Heterozygous *Cdc20^{fl/+}* mice were obtained by crossing the founder
mice and the wild type C57BL/6N mice. The genotype of F1 heterozygous *Cdc20^{fl/+}*

mice was confirmed by PCR amplification, DNA sequencing and Southern blot
analysis. *Sp7-Cre* mice were generated from Beijing Biocytogen Co., Ltd. To avoid
disrupting *Sp7* expression, iCre will be inserted between the coding sequence of
exon2 and 3'UTR. 2A will be used to achieve the expression of *Sp7* and iCre at the
same time and level. To avoid disrupting poly (A) signal of *Sp7* and promoter region
of the downstream gene *Aaas*, *Neo* cassette flanked by *frt* sites will be inserted within
the non-conserved region of intron. Genotyping primers: Osx-iCre-WT-F
TACCAGAAGCGACCAC-TTGAGC; iCre-Mut-R,
GCACACAGACAGGAGCATCTTC. Primers were showed in Table3. The detailed
information of the iB-Sp7-iCre mice was shown on this website:
<https://biocytogen.com/products/cre-mouse-rat-models/b-sp7-osx-icre-mice/>. This
Sp7-Cre (Osx-Cre) mouse model has been used to examine the effects of specific
gene loss in osteoblast progenitor cells on bone development (Chen *et al*, 2021; Jin *et*
*al*, 2017). The application of this mouse model largely verifies that the Sp7-Cre
transgene is able to specifically target Osx⁺ osteoprogenitors." The process mentioned
was shown in Fig EV1A. The genotypes of transgenic mice were shown in Fig EV1B.
The *Cdc20^{ff}* mice were in control groups, while *Sp7-Cre;Cdc20^{ff}* in experimental
groups. As for Osx-GFP::*Cre* (tet-off) model, the description is on this website:
<https://www.jax.org/strain/006361>. Cre-mediated recombination in these mice is
under the control of doxycycline. Thus, in the absence of doxycycline, EGFP-Cre
fusion protein expression is restricted in osteoblast lineage. A Sp7-tTA,
tetO-EGFP/Cre mouse model was generated by using the BAC of RP23-399N14

(204kb) in 2006. Since the BAC contains some other genes such as *Aaas*, *Myg1*,
 *Espl1* (except for *Sp7*), it results in a few potential risks of unexpected insertion of
 those genes, as well as the unclear position of the inserted BAC. While our strategy is
 to insert the Cre in a specific gene position to avoid generating the other unexpected
 transgene mice. Besides, the mouse model used in this research was lack of
 doxycycline regulation. The Cre recombinase-mediated removal of exon1-7 is
 expected to lead a translation termination of *Cdc20*. The two mouse models are
 available to conditional knockout specific gene targeting osteoprogenitors.

**Figure EV1A, B The manufacture and verification of conditional knockout**
 **mouse.**

(EV1A) The design strategy of conditional deletion of *Cdc20* gene. (EV1B) Representative image
 of PCR genotypes of indicated mice. *Sp7-Cre;Cdc20^{fl/fl}* mice were in experimental groups, *Cdc20^{fl/fl}*
 mice were in control groups.

2- Tail vein/systemic injections of lentivirus encoding shRNAs to knockdown gene
 expression in mice are not commonly performed. Although we understand that

lentiviruses were injected during 4 weeks in CDC20 CKO mice, we do not know how
much virus was inoculated, and how often. This information must be written in the
manuscript. Since lentiviruses are inoculated in a systemic fashion, various "side
effects" of p65 knockdown could be induced, and could complicate the interpretation
of the rescue experiment. Ideally p65 could be conditionally knocked-out of
osteoprogenitors, since floxed p65 mice exist. Alternatively, the authors should
provide evidence that mice inoculated with lentiviruses during 4 weeks are healthy,
and that apoptosis of liver hepatocytes is not induced in these mice, since whole-body
deficiency of p65 leads to massive liver apoptosis and death. p65 expression in the
liver of experimental and control animals should be evaluated, and perhaps in other
organs as well.

**Response:** Thanks for notification. Admittedly, injections of lentivirus are not the best
way, however, it can provide evidence in addition to our cellular results. The detailed
information was added in the manuscript page 25, line 546 as follows.

“For p65 lentivirus injections, 8×10^8 TU/ml lentiviruses 100 μ l every two weeks were
injected for twice via the tail vein. Four weeks later, the mice were sacrificed.” We
captured the pictures of the gross appearance of NC and p65 knockdown mice and
compared the body weight. The results showed that the NC and p65sh mice were
healthy, and there were no significant differences in the gross appearance and body
weight (Appendix Fig S7A, B). The knockdown efficiency of p65 in the liver of NC
and p65sh mice was determined by western blot assay (Appendix Fig S7C). Then we
tried to clarify the conditions of livers after lentivirus injection. According to

Appendix Fig S7D, there were no significant differences in H&E staining and
TUNEL staining, suggesting that apoptosis was not induced in p65sh mice. Some
researchers also used lentivirus injection to knockdown specific gene in the systemic
way (Krishnamachary *et al*, 2009; Luk *et al*, 2020).

**Appendix Fig S7A-D No significant adverse effects were found in tail vein**
**injection of p65sh mice and the efficiency of p65 knockdown in liver.**

(S7A, B) The appearance and body weight of NC and p65sh lentivirus injected mice. Scale bar,
1cm. Results are shown as mean \pm SD; (n=5); *ns*, *not significant*. (S7C) Western blot of p65
knockdown in the liver of NC and p65sh mice. (S7D) The H&E staining and TUNEL staining in
the liver of NC and p65sh mice. Scale bar: 50 μ m.

3- Additional controls and time-points should be presented in the manuscript in order
to better characterize the skeletal phenotype of the Cdc20 CKO mice. The bone
phenotype of Sp7-Cre; +/+ (WT) and Sp7-Cre; fl/+ (conditional HETs) must be
shown in addition to Cre-negative animals. Since Osterix is also expressed in growth
plate chondrocytes, it is important to characterize the growth plate phenotype of the
Sp7-Cre; fl/fl (CKO) mice. Part of the bone phenotype shown in 6-week-old mice
could result from delayed endochondral bone formation, due to altered cartilage
formation. It is important to know if this is the case. Characterization of endochondral
bone development in embryos and newborn mice should provide valuable information.
It would be interesting and important to provide data on the relative expression of
CDC20 and p65 (mRNA and/or protein) in chondrocytes, osteoprogenitors,
osteoblasts and osteocytes in wild type and CKO mice, on bone sections.

**Response:** Thanks for the important points. Following the referee's suggestion, we
investigated the skeletal phenotypes of the following groups including *Sp7-Cre*,
*Cdc20^{ff}*, *Sp7-Cre;Cdc20^{fl/+}* and *Sp7-Cre;Cdc20^{ff}*. Micro-CT analyses of the distal
femur metaphysis of *Sp7-Cre;Cdc20^{ff}* mice showed an impairment in trabecular bone
micro-architecture compared to other controls. There were no significant differences
among *Sp7-Cre*, *Cdc20^{ff}*, *Sp7-Cre;Cdc20^{fl/+}* phenotypes (Appendix Fig S1A). The
BMD and BV/TV parameters in *Sp7-Cre;Cdc20^{ff}* 6-week-old mice were greatly
lower than their littermates. *Sp7-Cre;Cdc20^{ff}* mice also had decreased Tb.N and
increased Tb.Sp while there were no significant differences among other control

groups (Appendix Fig S1B-E). The identification of *Sp7-Cre*, *Cdc20^{ff}*,
 *Sp7-Cre;Cdc20^{ff/+}* and *Sp7-Cre;Cdc20^{ff/ff}* genotypes were shown (Appendix Fig S1F).

**Appendix Fig S1A-F. Conditional knockout of Cdc20 impairs bone formation.**

(S1A) Representative micro-CT images of trabecular bone from the femoral metaphysis of

6-week-old *Sp7-Cre*, *Cdc20^{ff}*, *Sp7-Cre;Cdc20^{ff/+}* and *Sp7-Cre;Cdc20^{ff/ff}* mice. Scale bar, 500 μ m.

(S1B-E) Histomorphometric analyses of 6-week-old femurs. Results are shown as mean \pm SD;

(n=3); *ns*, not significant, ***p*<0.01, ****p*<0.001. (S1F) Representative image of PCR genotypes
of *Sp7-Cre*, *Cdc20^{ff}*, *Sp7-Cre;Cdc20^{ff/+}* and *Sp7-Cre;Cdc20^{ff/ff}* mice.

The growth plate phenotypes of 6-week-old mice were conducted to Goldner's
trichrome staining and the lengths of growth plates were evaluated through ImageJ
software. The results showed that there were no significant differences between the
lengths of growth plates of *Cdc20^{ff/ff}* and *Sp7-Cre;Cdc20^{ff/ff}* mice (Appendix Fig S2A,
B). As for characterization of endochondral bone development, we collected the
femurs and tibiae of embryonic day 19 (E19), postnatal day 1 (P1) and day 4 (P4)
mice and evaluated using H&E staining and Safranin-O-Fast Green staining. There
were no discernable differences among the endochondral bone development in
embryos and newborn mice (Appendix Fig S2C, D).

Fig S2A

Fig S2B

Fig S2C

Fig S2D

**Appendix Fig S2A-D The phenotypes of growth plates and endochondral bone**
**development.**

(S2A) Representative Goldner's trichrome staining of chondrocytes from the femoral metaphysis
of 6-week-old *Cdc20^{ff}* and *Sp7-Cre;Cdc20^{ff}* mice. Scale bar, 200 μ m. (S2B) Statistical analyses of
growth plate height of 6-week-old mice femurs. Results are shown as mean \pm SD; (n=6); *ns*, *not*
*significant*. (S2C, D) Representative H&E staining and Safranin-O-Fast Green staining of femurs
and tibia of embryonic day 19 (E19), postnatal day 1 (P1) and day 4 (P4) of *Cdc20^{ff}* and
*Sp7-Cre;Cdc20^{ff}* mice. Scale bar, 200 μ m.

Using immunofluorescence methods, we found the co-localization of CDC20 and p65
on bone sections (Appendix Fig S4E).

**Appendix Fig S4E The co-localization of CDC20 and p65 on bone sections**

The co-localization of CDC20 and p65 on bone sections of mice. Scale bar, 50 μ m.

During the osteoblast differentiation, BMSCs, osteoblasts and the precursor cells
could secrete transcription factors to form the bone extracellular matrix. And the
osteoprogenitors, osteoblasts, and osteocytes are all differentiated from BMSCs
(Komori, 2006). Therefore, it is hard to distinguish these types using specific markers
or particular morphology, and we used the markers osteocalcin (OCN), Runt-related
transcription factor 2 (RUNX2) to represent them. In conditional knockout mice, the
expression of CDC20, RUNX2 and OCN decreased, while p65 expression increased
(Appendix Fig S3F). As for chondrocytes, we used Collagen Type II(COL-2) to

characterize it. The results showed that the expression of COL-2 and CDC20 were
stable in the cartilage of *Cdc20^{ff}* and *Sp7-Cre;Cdc20^{ff}* mice (Appendix Fig S4F).

**Appendix Fig S3F Immunofluorescence of relative expression of CDC20, p65,**
**OCN, RUNX2 on bone sections**

Immunofluorescence of relative expression of CDC20, p65, OCN, RUNX2 on bone sections
of *Cdc20^{ff}* and *Sp7-Cre;Cdc20^{ff}* mice. Scale bar, 20 μ m.

**Appendix Fig S4F Immunofluorescence of relative expression of COL2 and**
**CDC20 on bone sections**

Immunofluorescence of relative expression of COL2 and CDC20 on bone sections of *Cdc20^{ff}* and
*Sp7-Cre;Cdc20^{ff}* mice. Scale bar, 20 μ m.

Minor comments:

1- Histological analyses of the scaffold engrafted subcutaneously in nude mice (Fig.
1J, and 1K) should be further analyzed by in situ hybridization with osteoblast
markers (such as Collagen Type I and/or osteopontin), to support the conclusion that
CDC20 knockdown in hBMDCs impairs osteoblast differentiation

**Response:** Thanks for reminding. As suggested, we aimed to figure out the relative
mRNA and/ or protein levels of osteoblast markers, and we used the
Immunohistochemistry method. The results showed that the expression of osteocalcin
(OCN) and osteopontin (OPN) decreased in CDC20sh scaffold, suggesting that
knockdown of CDC20 in hBMSCs impaired osteoprogenitor differentiation. The
results were shown in Appendix Fig S2F. The description was added in the manuscript
page 9 line 197 as follows.

“The expression of osteoblast markers osteocalcin (OCN) and osteopontin (OPN)
decreased in CDC20sh scaffold using IHC staining (Appendix Fig S2F). These results
showed that knockdown of CDC20 impaired osteogenic differentiation of BMSCs.”

Fig S2F

**Appendix Fig S2F. The OCN and OPN expression in NC and CDC20sh scaffold.**

The expression of OCN and OPN in sections of NC and CDC20sh scaffolds using IHC assay.

Scale bar, 50 μ m.

2- Fig 2E, indicates that the holes in tibiae of control and CKO have been drilled on
opposite sides of the bone. Since the cortical thickness varies depending on the side of
the tibia, it is important to generate bone defects at the exact same location for all
control and CKO mice. It is also important to specify at which age these defects have
been created.

**Response:** Thanks for reminding. As suggested, we have repeated the experiments
and evaluated the position of the holes. The description has been added in the
manuscript page 33 line 708 as follows.

“For tibial cortical bone defect, the 0.8 mm diameter holes were produced on the
anterior surface of the tibiae with a round bur (Komet, Germany) conducting at 1000
rpm with saline irrigation at about 2mm under the knees. These defects were
conducted on 12-week-old mice, and these mice were sacrificed ten days later.”

Results were presented in Fig 1E, F.

**Fig 1E, F Representative micro-CT images and histomorphometric analysis of**
**the regenerated bone in tibial cortical gaps**

(1E) Representative micro-CT images of tibial cortical bone defects in *Sp7-Cre;Cdc20^{fl/fl}* and
littermate control mice. The green lines show the position of the original defect position. The
green circle represents the regenerated bone. Scale bar, 500 μm. (1F) Histomorphometric analysis
of the regenerated bone in tibial cortical gaps. Results are shown as mean ± SD; (n=5);
*** $p < 0.001$.

3- Fig. 6F, 3rd lane: there is a large APC11 band, while the legend indicates no
HA-APC11 overexpressed in this sample. The legend appears correct since p65
protein is upregulated in this sample. The APC11 blot should show a band in the
second lane, not the third lane. This implies that immunostaining of p65 and of
APC11 has been done on two unrelated blots. The entire experiment should be done
again using the same membrane to detect p65, APC11, CDC20 and GAPDH.

**Response:** Thanks for reminding. As suggested, we repeated experiments and
re-presented the figure. During the experiments, we adjusted the sequences of groups
and misused the former lanes. We have re-evaluated it and changed the whole figure.

Fig 6F

**Fig 6F Western blot analyses of p65 expression in NC or CDC20sh HEK293T**
**cells after the overexpression of Vector or HA-APC11 plasmids.**

4- Fig. 7C shows robust induction of p65 ubiquitination in vitro in presence of
purified CDC20 without APC11. Does this mean that APC11 is not required for
CDC20-induced ubiquitination of p65?

**Response:** Thanks for suggestions. Among the in vitro ubiquitination assay, the
substrates Flag-p65 were obtained from cell lysis, the interaction of APC11 and p65
were illustrated above and APC11 existed in the system, so it can not exclude the
influence of APC11 in the induction of p65 ubiquitination. Some researchers also use
this method to investigate the in vitro ubiquitination, the purified GST-tagged proteins
from bacterial and substrates from cells were involved (Liu *et al*, 2016; Wang *et al*,
2016; Wei *et al*, 2017). In addition, we added the in vitro ubiquitination promoted by

GST-CDC20 or GST-CDC20 with GST-APC11. The combination of GST-CDC20
 with GST-APC11 have exerted more ubiquitination of p65 than GST-CDC20 alone,
 illustrating APC11 can enhance the effect of CDC20 but not required. Results were
 shown in Appendix Fig S5A.

**Appendix Fig S5A Immunoblot of Flag-p65-linked in vitro ubiquitination**
 **promoted by GST-CDC20 or GST-CDC20 with GST-APC11.**

Bacterially expressed and purified GST-CDC20 or GST-CDC20 with GST-APC11 were incubated
 with purified proteins, including HA-ubiquitin, E1, E2, and Flag-p65 at 32 °C for 1 h.

5- Fig 7F, middle lane shows a strong downregulation of CDC20 protein expression
 although no shRNAs against CDC20 are expressed. How do you explain this?

**Response:** Thanks for notification. As suggested, we re-evaluated the lanes and
 repeated the experiments. We used CDC20sh HEK293T cells with high knockdown
 efficiency and found the expression of CDC20 in middle lane did not decrease
 compared to the first lane, and we have changed the whole figure of Fig 7F.

**Fig 7F Immunoblot of Flag-p65-linked ubiquitination promoted by HA-APC11**
 **in NC and CDC20sh cells.**

NC or CDC20sh HEK293T cells were co-transfected with Flag-p65, His-Ubiquitin,
 with or without HA-APC11 and Vector plasmids. Transfected cells were treated with
 10 μ M MG132 for 6 h before collection.

6- The presentation of the results could be slightly reorganized by presenting first the
 mouse model and its skeletal phenotype, and then in vitro experiments with BMDCs
 (currently presented in Fig. 1). Moreover, the presentation of interaction of CDC20
 and p65 both in the text and in the figures appears somewhat lengthy and could be
 shortened. Finally, the results presented in fig. 6I-L could be transferred to figure 8,
 since they support the conclusion that p65/NF κ B signaling plays a role downstream of
 CDC20 in osteoprogenitor differentiation.

**Response:** Thanks for the reminding. As suggested, we have reorganized the figure.
We have changed the sequences of Fig 1 and Fig 2. The interaction of CDC20 and
p65 in text has been shortened, the related figures in Fig EV4 have been transformed
in Appendix Fig S4. The results presented in Fig 6I-L have been changed in Fig 8E, F
and Fig 5EVF, G.

7- The protocol used to generate BMSCs is unclear.

**Response:** Thanks for the notification. As suggested, we have supplemented the
description of generating BMSCs in page 22 line 476 as follows.

“Mice were sacrificed and femurs and tibiae were collected. Small cuts
(approximately 1 to 2 mm) were made at both the proximal and distal ends of the
bones and bone marrow was flushed into the collection tubes. Then bones were cut up
and digested in collagenase type 2 and type 4 (15 mg/ml, Worthington Biochemical
Corporation) for 45 min. The supernatants of the digestion were mixed together with
the flushed bone marrow for centrifugation, and the cell pellets were placed into the
suitable dishes. After 48 h of culture, media was removed, aspirated and non-adherent
cells were discarded. The cells were digested and the cell pellets were resuspended in
the volume of culture media for plating. When the BMSCs reached confluency, they
were proceeded to perform differentiation (Maridas *et al*, 2018).”

Very minor additional comments:

1- The term "in vivo" page 5, line 108 is not appropriate since it refers to in vitro
experiments.

**Response:** As suggested, we have corrected them in the manuscript. The term "in
vivo" in page 5, line 108 has been deleted in page 8, line 171 as follows.

"To further investigate the important role of CDC20 in osteogenic differentiation of
BMSCs, we gathered BMSCs from *Cdc20^{ff}* control mice and *Sp7-Cre;Cdc20^{ff}*
experimental mice through flushing bone marrow and digesting bone tissues with
collagenase."

2- Page 6, line 121 the text "decreased ALP staining and activity" could be replaced
by "decreased ALP staining and quantification" since the staining actually reflects
ALP activity.

**Response:** The text "decreased ALP staining and activity" in page 6, line 121 has
been changed to "decreased ALP staining and quantification" in page 9, line 184 as
follows. "Moreover, the CDC20 knockdown hBMSCs showed decreased ALP
staining and quantification (Fig 2E, F)."

3- The term "in vivo" page 13 line 282 may be misleading, since this is not observed
in live mice but in cells cultured in vitro. This could be replaced by "in cells".

**Response:** The term "in vivo" in page 13 line 282 has been replaced by "in cells" in
page 16 line 332 as follows.

"These results support p65 as a direct APC11^{CDC20} substrate in cells and in vitro."

Referee #2:

The authors describe a novel function of CDC20, a co-activator of the E3 ubiquitin
ligase APC/C, in bone remodeling and regeneration. Whereas in vivo data to
demonstrate this function were obtained by the generation and analysis of mice with
Sp7-Cre-mediated CDC20 inactivation in bone-forming osteoblasts, the majority of
experiments were set up to address the molecular bases explaining the deduced
function of CDC20 in bone formation. Overall, this is a very complex study with
strong and potentially relevant data. There are however several issues that remain to
be addressed.

Specific comments:

1) Although the reported results are surely relevant from a basic science perspective,
the statement that APC11CDC20 is a novel target for osteoporosis treatment appears
too strong. Is it truly realistic to establish an agonist of such a complex cytoplasmatic
regulator, whose expression is not restricted to osteoblasts?

**Response:** Thanks for the important points. From our results above, loss of CDC20 in
osteoprogenitors impaired bone formation, overexpression of CDC20 improved the
osteogenic differentiation of MSCs. Targeting APC11^{CDC20} may influence the
functions of osteoprogenitors, more evidence on its application should be investigated
further. Perhaps we can change it to “Our current work clarified a cell-cycle

independent function of CDC20, establishing APC11^{CDC20} as a pivotal regulator for
bone formation by governing the ubiquitination and degradation of p65, and may
provide a novel clue in the treatment of bone-related diseases.” in page 2, line 35.

2) The authors have to provide more information about their Sp7-Cre model, which
apparently has been generated for this specific project. The key question is, if these
mice represent a Cre knockin into the Sp7 locus, which might interfere with the
endogenous expression of the relevant transcription factor Osterix. If this is the case,
the Cdc20f/f mice alone would not be sufficient as a control, and it has to be
evaluated if the Sp7-Cre mice display a skeletal phenotype, regardless of their Cdc20
genotype.

**Response:** Thanks for the constructive points. Osterix (Osx, Sp7) is a specific
transcription factor of osteoblast precursor cells and we generated *Sp7-Cre;Cdc20^{ff}*
mice to examine the role of CDC20 during osteoblast precursor differentiation and
bone formation. We have added the description about the mouse model in the methods
and materials in manuscript in page 23, line 493 as follows.

“*Cdc20^{ff}* and *Sp7-Cre;Cdc20^{ff}* mice were generated by Biocytogen Co., Ltd. (Beijing,
China) using CRISPR/Cas9 based EGE system. Briefly, single guide RNAs (sgRNAs)
targeting the upstream of exon1 and the downstream of exon7 of *Cdc20*, respectively,
were designed using the CRISPR design tool (<http://www.sanger.ac.uk/htgt/wge/>).
The candidate sgRNAs were screened for on-target activity using the UCA™
CRISPR efficiency evaluation kit (Biocytogen Co., Ltd., Beijing, China), and two

sgRNAs with high specificity and on-target activity were selected. The targeting
vector that contains the genomic DNA spanning exon1-7 of mouse *Cdc20* flanked by
two loxP sites and the homology arms at the 5' and 3' regions were constructed as
well. The targeting vector, the in vitro-synthesized sgRNAs and Cas9 mRNA were
co-injected into C57BL/6N mouse zygotes. After injection, the surviving 2-cell-stage
zygotes were transplanted into the KM albino pseudopregnant females. The founder
mice bearing the floxed *Cdc20* allele were determined by PCR amplification and
DNA sequencing. Heterozygous *Cdc20*^{fl/+} mice were obtained by crossing the founder
mice and the wild type C57BL/6N mice. The genotype of F1 heterozygous *Cdc20*^{fl/+}
mice was confirmed by PCR amplification, DNA sequencing and Southern blot
analysis. *Sp7-Cre* mice were generated from Beijing Biocytogen Co., Ltd. To avoid
disrupting *Sp7* expression, iCre will be inserted between the coding sequence of
exon2 and 3'UTR. 2A will be used to achieve the expression of *Sp7* and iCre at the
same time and level. To avoid disrupting poly (A) signal of *Sp7* and promoter region
of the downstream gene *Aaas*, *Neo* cassette flanked by *frt* sites will be inserted within
the non-conserved region of intron. Genotyping primers: Osx-iCre-WT-F
TACCAGAAGCGACCAC-TTGAGC; iCre-Mut-R,
GCACACAGACAGGAGCATCTTC. Primers were showed in Table3. The detailed
information of the iB-Sp7-iCre mice was shown on this website:
<https://biocytogen.com/products/cre-mouse-rat-models/b-sp7-osx-icre-mice/>. This
*Sp7-Cre* (*Osx-Cre*) mouse model has been used to examine the effects of specific
gene loss in osteoblast progenitor cells on bone development (Chen *et al*, 2021; Jin *et*

*al*, 2017). The application of this mouse model largely verifies that the Sp7-Cre
transgene is able to specifically target Osx+ osteoprogenitors.”

Following the referee’s suggestion, we investigate the skeletal phenotype of the
following groups including *Sp7-Cre*, *Cdc20^{ff}*, *Sp7-Cre; Cdc20^{ff/+}* and *Sp7-Cre;*
*Cdc20^{ff}*. Micro-CT analyses of the distal femur metaphysis of *Sp7-Cre; Cdc20^{ff}* mice
showed an impairment in trabecular bone micro-architecture compared to other
controls. There were no significant differences among *Sp7-Cre*, *Cdc20^{ff}*, *Sp7-Cre;*
*Cdc20^{ff/+}* phenotypes (Appendix Fig S1A). The BMD and BV/TV parameters in
*Sp7-Cre;Cdc20^{ff}* 6-week-old male mice were greatly lower than their littermates.
*Sp7-Cre;Cdc20^{ff}* mice also had decreased Tb.N, and increased Tb.Sp while there were
no significant differences among other control groups (Appendix Fig S1B-E). The
identification of *Sp7-Cre*, *Cdc20^{ff}*, *Sp7-Cre; Cdc20^{ff/+}* and *Sp7-Cre; Cdc20^{ff}* genotypes
were shown in Appendix Fig S1F.

**Appendix Fig S1A-F. Conditional knockout of Cdc20 impairs bone formation.**

(S1A) Representative micro-CT images of trabecular bone from the femoral metaphysis of

6-week-old *Sp7-Cre*, *Cdc20^{fl/fl}*, *Sp7-Cre;Cdc20^{fl/+}* and *Sp7-Cre;Cdc20^{fl/fl}* mice. Scale bar, 500 μ m.

(S1B-E) Histomorphometric analyses of 6-week-old femurs. Results are shown as mean \pm SD;

(n=3); ns, not significant, ** p <0.01, *** p <0.001. (S1F) Representative image of PCR genotypes

of *Sp7-Cre*, *Cdc20^{fl/fl}*, *Sp7-Cre;Cdc20^{fl/+}* and *Sp7-Cre;Cdc20^{fl/fl}* mice.

3) Since the in vivo findings in *Sp7-Cre/Cdc20^{f/f}* mice are essentially the basis for the
molecular experiments described thereafter, it would be important to strengthen them.
In fact, there was no analysis performed to understand the cellular causes of the
osteopenia. This could be done by cellular histomorphometry quantifying osteoblast
and osteoclast numbers on the femur sections. Alternatively, the authors could
measure serum levels of bone formation and resorption biomarkers. As presented, the
phenotype description is incomplete.

**Response:** Thanks for suggestions. We used markers osteocalcin (OCN), Collagen
Type I (COL1) to characterize osteoblasts, and Tartrate-resistant acid phosphatase
(TRAP) staining to characterize osteoclasts using IHC assay (Appendix Fig S3A). We
calculated the numbers of positive expression cells per perimeter of bone matrix. The
measurement of osteoblasts and osteoclasts were calculated using ImageJ software
and presented as N.Ob/B.Pm (osteoblast number/bone perimeter) or N.Oc/B.Pm
(osteoclast number/bone perimeter) and no significant differences were seen in
*Cdc20^{f/f}* and *Sp7-Cre;Cdc20^{f/f}* mice (Appendix Fig S3B). Additionally, the serum
levels of PINP and CTX-1 were measured through Elisa assay. The results showed
that the bone formation biomarker PINP decreased in *Sp7-Cre;Cdc20^{f/f}* mice, while
bone resorption marker CTX-1 did not change (Appendix Fig S3D, E). The results
implied that the decreased function of osteoblasts was not due to the differences of
cell numbers. Perhaps the CDC20 loss in osteoprogenitors resulting in activation of
p65 may provide some evidence.

**Appendix Fig S3A, B, D, E The numbers and the expression of biomarkers of**
 **osteoblasts and osteoclasts.**

(S3A) Immunohistochemistry of OCN and COL1, and TRAP staining of 6-week-old *Cdc20^{ff}*,
 *Sp7-Cre;Cdc20^{ff}* mice. The arrows represent the positive expression cells. Scale bar, 50 μ m. (S3B)
 Measurements and statistical analyses of numbers of osteoblasts and osteoclasts in bone sections
 of *Cdc20^{ff}* and *Sp7-Cre;Cdc20^{ff}* mice. (n=6) (S3D, E) Measurements and statistical analyses of
 PINP and CTX-1 in the serum of *Cdc20^{ff}* and *Sp7-Cre;Cdc20^{ff}* mice. (n=6) Results are shown as
 mean \pm SD; ns, not significant, * p <0.05. N.Ob/B.Pm (osteoblast number/bone perimeter),
 471 N.Oc/B.Pm (osteoclast number/bone perimeter).

4) The molecular experiments shown in the second part of the manuscript are overall
convincing, although I personally consider the data obtained in BMCSs (now mostly
shown as Supplemental Figures) as more relevant than the data obtained in HEK cells.
Moreover, although some of the Western blot results are confirmed by quantitative
data, many others are not. Since the authors state that all data were collected from at
least three independent experiments, it should be possible to perform quantification
and statistical analysis for all presented data.

**Response:** Thanks for notifications. In the mechanical experiments, we mainly use
HEK293T cells. Lots of immunoprecipitations as well as ubiquitination experiments
were involved in our research, which requires large numbers of cells and high
efficiency of transfection. Compared to BMSCs, HEK293T cells are easier to
transfect and are able to produce large amounts of recombinant proteins. We have
verified the important experiments in BMSCs but not the whole. Therefore, we
choose to show results from HEK293T in the main figure related to the mechanical
experiments and we have changed the degradation of p65 by CDC20 in BMSCs to the
main figure. Furthermore, we have repeated the experiments at least three times and
we provide quantifications and statistical analyses of the significant differences in
western blot experiments in Appendix Fig S8.

**Appendix Fig S8 Quantifications and statistical analyses of the significant**
 **differences in western blot experiments.**

Results are shown as mean \pm SD; $n \geq 3$; * $p < 0.05$; ** $p < 0.01$; *** $p < 0.001$.

5) In the last paragraph of the Results section the authors state that "..., lentiviruses

were distributed in mice and the expression levels of p65 was knocked down

according to qRT-PCR and western blot analyses." While this statement suggests that
p65 expression was analyzed in vivo, the legend of Figure EV7B/C indicates that
these results were obtained in BMSCs. If this means that the expression analysis was
performed in ex vivo cultures, this has to be clearly stated.

**Response:** Thanks for reminding. As suggested, we have corrected it in page 17, line
367 as follows.

"To clarify the efficiency of p65 knockdown in bone sections, lentivirus was
distributed in mice (Fig EV5A), the expression levels of p65 in BMSCs obtained from
femurs and tibiae were examined according to qRT-PCR and western blot analyses
(Fig EV5B, C)."

6) Given the complexity of their findings, the authors should think about providing a
schematic presentation to illustrate their conclusions.

**Response:** Thanks for suggestions. We have provided the schematic figure
concerning about the mechanisms and the functional experiments in cells and in mice
in Appendix Fig S7E. The left model shows the relationship of APC11, CDC20, p65
and ubiquitin. The WD40 domain of CDC20 interacts with the RHD domain of p65,
which subsequently transfers polyubiquitin to p65 inducing its degradation in a
proteasome-dependent manner and promoted the osteogenesis of BMSCs through
NF- κ B pathway. APC11 is involved in this process. The right model shows the
method to generate the conditional knockout mice and loss of Cdc20 in
osteoprogenitors in mice results in decreased bone formation.

**Appendix Fig S7E. A model depicting how CDC20 regulates bone formation**
 **through p65.**

The left model shows the relationship of APC11, CDC20, p65 and ubiquitin. The right model
 shows the method to generate the mice and loss of Cdc20 in osteoprogenitors in mice results in
 decreased bone formation.

Referee #3:

In this manuscript, Du et al. report a positive regulation of osteogenesis by the
 APC/C(Cdc20) ubiquitin E3 ligase complex through modulating p65/RelA stability.

The authors observed a positive correlation between Cdc20 expression and
 osteogenesis using both in vitro BMSC culture/differentiation models and an in vivo

conditional Cdc20 knockout mouse model. There is a decreased bone mass found in
 Cdc20-depleted animals. The authors then sought to define the underlying mechanism,

they found p65 was upregulated in Cdc20-deficient BMSCs, hence hypothesized p65
 as an APC(Cdc20) ubiquitin substrate. They provided evidence that p65 bound to

Cdc20 and that Cdc20 facilitates p65 ubiquitination. To assess the role of p65 in

Cdc20-mediated osteogenesis, the authors delivered p65sh lentivirus into the mice
and were able to observe a rescued phenotype. This is an interesting study
demonstrating a novel role of Cdc20 in osteogenesis. However, the data in their
current form are quite preliminary, hence a major revision is recommended to provide
strong evidence to support the conclusions. Below are specific comments that should
be addressed prior to being considered for publication in EMBO Reports.

1. Cdc20 is essential for M->G1 transition, complete loss of Cdc20 arrests cell cycle
in M phase and eventually causes cell death. Can the authors comment on whether the
phenotypes observed are at least in part due to decreased osteoblast numbers upon
*Osx/Sp7*-driven deletion of Cdc20?

**Response:** Thanks for the critical comments. As suggested, we used markers
osteocalcin (OCN), Collagen Type I (COL1) to characterize osteoblasts using IHC
assay (Appendix Fig S3A). The numbers of osteoblasts were calculated using ImageJ
software and presented as N.Ob/B.Pm (osteoblast number/bone perimeter) and no
significant differences were seen in *Cdc20^{ff}* and *Sp7-Cre;Cdc20^{ff}* mice (Appendix Fig
S3B). Additionally, the EdU assay conducted on mice showed no discernable
differences of proliferation ability between *Cdc20^{ff}* and *Sp7-Cre;Cdc20^{ff}* mice
(Appendix Fig S3C). The results implied that the decreased function of osteoblasts
was not due to the differences of cell numbers. Perhaps the phenotype of bone loss in
*Sp7-Cre;Cdc20^{ff}* mice was mainly due to the reduced function of osteoblasts, and the

CDC20 loss in osteoprogenitors resulting in activation of p65 may provide some
evidence.

**Appendix Fig S3A-C. The measurements of numbers of osteoblasts and the**
**proliferating cells.**

(S3A) Immunohistochemistry of OCN and COL1 of 6-week-old *Cdc20^{ff}*, *Sp7-Cre;Cdc20^{ff}* mice.

The arrows represent the positive expression cells. Scale bar, 50 μ m. (S3B) Measurements and

statistical analyses of numbers of osteoblasts in bone sections of *Cdc20^{ff}* and *Sp7-Cre;Cdc20^{ff}*

mice. (n=6) Results are shown as mean \pm SD; *ns*, *not significant*. (S3C) Immunofluorescence of

EdU of *Cdc20^{ff}*, *Sp7-Cre;Cdc20^{ff}* mice femurs. Scale bar, 50 μ m. N.Ob/B.Pm (osteoblast

number/bone perimeter).

2. Fig. 1H-I, can the authors comment on why the Cdc20 WD40 domain was able to

rescue the decreased ALP activity upon Cdc20 loss?

**Response:** Thanks for the points. WD40 domain of CDC20 was reported to be
engaged in multiple protein-protein interactions. CDC20 appeared to bridge the
interactions between APC/C and the substrates through this structure (Yu, 2007). As
our results showed above, WD40 domain of CDC20 was responsible for the
interaction and degradation of p65, thus influencing the osteogenesis of BMSCs.
Therefore, overexpression of WD40 domain of CDC20 can rescue the decreased ALP
activity and bone formation by inducing the ubiquitination of p65.

3. Fig. 2, Did the authors perform whole-mount skeleton staining of control and
Sp7-cre;Cdc20(f/f) mice?

**Response:** Thanks for reminding. As suggested, we have performed whole-mount
skeleton staining of *Cdc20^{ff}* and *Sp7-Cre;Cdc20^{ff}* postnatal day 2 (P2) mice. The
whole skeleton, upper extremities and hind limbs were presented (Appendix Fig S2E).
No significant differences were seen in the staining of skeleton.

Fig S2E

**Appendix Fig S2E. Whole-mount skeleton staining of *Cdc20^{ff}* and**
***Sp7-Cre;Cdc20^{ff}* P2 mice.**

The upper one shows the whole skeleton. The middle one shows the upper extremities,
and the lower one shows the hind limbs of *Cdc20^{ff}* and *Sp7-Cre;Cdc20^{ff}* mice. Scale
592 bar, 1cm.

Fig. 2, Have the authors tried co-staining Cdc20, p65 and an osteoblast marker using
immunohistochemistry?

**Response:** Thanks for suggestions. Using immunochemistry methods, we found the
co-localization of CDC20 and p65 on bone sections (Appendix Fig S4E). As for
osteogenic markers, we chose the early and late osteoblast marker, Runt-related
transcription factor 2 (RUNX2) and osteocalcin (OCN), to characterize the osteogenic
ability. In conditional knockout mice, the expression of CDC20, RUNX2 and OCN
decreased, while p65 expression increased (Appendix Fig S3F), suggesting that loss
of CDC20 in osteoprogenitors impaired the osteogenic differentiation.

Fig S4E

**Appendix Fig S4E The co-localization of CDC20 and p65 on bone sections**

The co-localization of CDC20 and p65 on bone sections of mice. Scale bar, 50 μ m.

Fig S3F

**Appendix Fig S3F Immunofluorescence of relative expression of CDC20, p65,**

**OCN, RUNX2 of *Cdc20^{ff}* and *Sp7-Cre;Cdc20^{ff}* mice on bone sections.**

Immunofluorescence of relative expression of CDC20, p65, OCN, RUNX2 on bone sections

of *Cdc20^{ff}* and *Sp7-Cre;Cdc20^{ff}* mice. Scale bar, 20 μ m.

5. Fig. 3, Did the authors check p-p65 levels in conditions where Cdc20 is depleted?

**Response:** Thanks for reminding. As suggested, we used shRNA to deplete CDC20

and conducted western blot assay. The results showed that the expression of p-p65

increased in CDC20sh hBMSCs (Appendix Fig S4B).

**Appendix Fig S4B Western blot analyses of p65, p-p65 in NC and CDC20sh**

**hBMSCs.**

6. Fig. 3, In siCdc20 cells, was p65 mRNA affected as well?

**Response:** Thanks for notification. As suggested, we conducted qRT-PCR
experiments in siCDC20 cells. The results showed that the expression of NF-κB
pathway downstream factor, *IL-6* and *IL-8* increased, while the expression of *p65*
remained stable (Appendix Fig S4A).

**Appendix Fig S4A The expression of *CDC20*, *p65* and NF-κB pathway**
**downstream genes *IL-6*, *IL-8* of NC and CDC20si hBMSCs determined by**
**qRT-PCR.**

Results are shown as mean ± SD; (n=5); ns, not significant, ***p<0.001.

7. Fig. 3 and 4, Why did the authors choose to focus on the p65/NF-κB pathway in
this study? Are other NF-κB pathway proteins found increased upon Cdc20 deletion
or interacted with Cdc20?

**Response:** Thanks for suggestions. Actually, we first found that in CDC20 deleted
cells, the osteogenesis of BMSCs decreased largely, so we tried to find the pathway
related to bone formation. Interestingly, we found the mRNA expression of NF-κB
pathway downstream factor significantly increased in CDC20si cells. Then we sought

to find the key protein and we found the interaction and degradation of CDC20 on
p65, so we chose to target on p65/NF- κ B pathway. As suggested, we evaluated the
interaction of other proteins in NF- κ B pathway with CDC20. Our results illustrated
that no interaction was found between CDC20 and I κ B α (Appendix Fig S 4C, D).
Based on the result, we clarified that the ubiquitination of p65 controlled by CDC20
influenced the osteogenesis of BMSCs.

**Appendix Fig S4C, D No interaction was found between CDC20 and I κ B α in**
**hBMSCs.**

8. Fig. 6, APC2 is required for the integrity of APC/C complex, did the authors
suggest in this study an APC2-absent but APC11-present APC/C subcomplex to
catalyze p65 ubiquitination? The authors should examine known APC(Cdc20)
substrates to demonstrate an effective depletion of APC2, APC11, and Cdh1.

**Response:** Thanks for notification. As suggested, we performed the ubiquitination
assay in NC and APC2sh HEK293T cells with the overexpression of APC11, and
found no significant differences of p65 ubiquitination (Appendix Fig S6A).
Researches have clarified the known substrate Cyclin B1 of APC11, APC2, CDH1
and other APC/C subunits (Almeida, 2012; Dimova *et al*, 2012; Pflieger *et al*, 2001).

Therefore, we chose Cyclin B1 as the known substrate to demonstrate the effective
 depletion of APC2, APC11 and CDH1 (Appendix Fig S6B-D). Our results implied
 that APC2 did not influence the ubiquitination of p65 by APC11. Some researchers
 have found the ability of APC11 to catalyze substrates without APC2 (Leverson *et al*,
 2000), demonstrating the catalytic ability of APC11 alone.

**Appendix Fig S6A-D Immunoblot of Flag-p65-linked ubiquitination promoted**

by HA-APC11 in NC and APC2sh cell and immunoblot of Cyclin B1-linked
ubiquitination after the knockdown of APC2, APC11, CDH1.

(S6A) NC or APC2sh HEK293T cells were co-transfected with Flag-p65,
His-Ubiquitin, with or without HA-APC11 and Vector plasmids. Transfected cells
were treated with 10 μ M MG132 for 6 h before collection. (S6B-D) NC and APC2sh,
APC11sh, CDH1sh HEK293T cells were treated with 10 μ M MG132 for 6 h, whole
cell protein extracts were immunoprecipitated with anti-Cyclin B1 and
immunoprecipitations were immunoblotted for Ubiquitin.

9. Fig. EV5, was Cdc20 upregulated when knocking down Cdh1?

**Response:** Thanks for reminding. As suggested, we used shRNA to knock down
CDH1 and found no obvious change of CDC20 expression in western blot (Fig
EV4F).

Fig EV4F

**Fig EV4F Western blot analyses of the expression of CDC20 in NC and CDH1sh**
**HEK293T cells.**

10. Fig. 7C, bacterially expressed and purified GST-Cdc20 and GST-APC11 are
APC/C-free, the authors need to explain why APC-free GST-Cdc20 and GST-APC11
promote the ubiquitination of p65 in vitro.

**Response:** Thanks for notification. Among in vitro ubiquitination assay, the
GST-tagged protein was gathered from bacterial, while FLAG-p65 was obtained from
cell lysis. The interaction of APC11 and p65 were illustrated above and APC11
existed in the system. The cell lysis contains the endogenous APC/C and intrigues the
ubiquitination of p65. Some researchers also use this method to investigate the in vitro
ubiquitination, the purified GST-tagged proteins from bacterial and substrates from
cells were involved (Liu *et al*, 2016; Wang *et al*, 2016; Wei *et al*, 2017). In our results,
we saw dramatic increase of ubiquitination of p65 added with GST-CDC20 or
GST-APC11, the combination of GST-CDC20 with GST-APC11 exerted more
ubiquitination of p65 than GST-CDC20, suggesting both enzymes can catalyze this
reaction and APC11 enhanced the ubiquitination of p65 by CDC20. Results were
shown in Appendix Fig S5A.

Fig S5A

**Appendix Fig S5A Immunoblot of Flag-p65-linked in vitro ubiquitination**
**promoted by GST-CDC20 or GST-CDC20 with GST-APC11.**

Bacterially expressed and purified GST-CDC20 or GST-CDC20 with GST-APC11
were incubated with purified proteins, including HA-ubiquitin, E1, E2, and Flag-p65
at 32 °C for 1 h.

Reference:

Almeida A (2012) Regulation of APC/C-Cdh1 and Its Function in Neuronal Survival. *MOL*
*NEUROBIOL* 46: 547-554

Chen Y, Fan Q, Zhang H, Tao D, Wang Y, Yue R, Sun Y (2021) Lineage tracing of cells expressing the
ciliary gene IFT140 during bone development. *Dev Dyn* 250: 574-583

Dimova NV, Hathaway NA, Lee BH, Kirkpatrick DS, Berkowitz ML, Gygi SP, Finley D, King RW
(2012) APC/C-mediated multiple monoubiquitination provides an alternative degradation signal for
cyclin B1. *Nat Cell Biol* 14: 168-176

Jin L, Chao L, Guo B, Wu X, Zhang G (2017) Increased PLEKHO1 within osteoblasts suppresses
Smad-dependent BMP signaling to inhibit bone formation during aging. *Aging cell* 16: 360-376

Komori T (2006) Regulation of osteoblast differentiation by transcription factors. *J Cell Biochem* 99:
1233-1239

Krishnamachary B, Glunde K, Wildes F, Mori N, Takagi T, Raman V, Bhujwala ZM (2009)
Noninvasive detection of lentiviral-mediated choline kinase targeting in a human breast cancer
xenograft. *Cancer Res* 69: 3464-3471

Levenson JD, Joazeiro C, Page AM, Huang HK, Hieter P, Hunter T (2000) The APC11 RING-H2

Finger Mediates E2-Dependent Ubiquitination. *Mol Biol Cell* 11: 2315-2325

Liu J, Wan L, Liu J, Yuan Z, Zhang J, Guo J, Malumbres M, Liu J, Zou W, Wei W (2016) Cdh1 inhibits
WWP2-mediated ubiquitination of PTEN to suppress tumorigenesis in an APC-independent manner.
*Cell Discov* 2: 15044

Luk S, Ng K, Zhou L, Tong M, Wong T, Yu H, Lo C, Man K, Guan X, Lee T (2020) Deficiency in
embryonic stem cell marker reduced expression 1 activates mitogen-activated protein kinase kinase
6-dependent p38 mitogen-activated protein kinase signaling to drive hepatocarcinogenesis. *Hepatology*
72: 183-197

Maridas D, Rendina-Ruedy E, Le PT, Rosen CJ (2018) Isolation, Culture, and Differentiation of Bone
Marrow Stromal Cells and Osteoclast Progenitors from Mice. *J VIS EXP* 2018: 131

Pflieger CM, Lee E, Kirschner MW (2001) Substrate recognition by the Cdc20 and Cdh1 components
of the anaphase-promoting complex. *Genes Dev* 15: 2396-2407

Wang Y, Zhang N, Zhang L, Li R, Fu W, Ma K, Li X, Wang L, Wang J, Zhang H et al (2016)
Autophagy Regulates Chromatin Ubiquitination in DNA Damage Response through Elimination of
SQSTM1/p62. *Mol Cell* 63: 34-48

Wei X, Wang X, Zhan J, Chen Y, Fang W, Zhang L, Zhang H (2017) Smurf1 inhibits integrin activation
by controlling Kindlin-2 ubiquitination and degradation. *J Cell Biol* 216: 1455-1471

Yu H (2007) Cdc20: a WD40 activator for a cell cycle degradation machine. *Mol cell* 27: 3-16

Dear Dr. Zhou,

Thank you for submitting your revised manuscript. It has now been seen by all of the original referees.

As you can see, the referee finds that the study is significantly improved during revision and recommends publication. However, referee #1 finds that the characterization of Sp7-iCre line is currently insufficient. We agree with referee #1 that addressing this concern would significantly strengthen the manuscript. However, being unable to address this point will not preclude from publication here.

Moreover, I need you to address the editorial points below before I can accept the manuscript.

- As per our format requirements, in the reference list, citations should be listed in alphabetical order and then chronologically, with the authors' surnames and initials inverted; where there are more than 10 authors on a paper, 10 will be listed, followed by 'et al.'. Please see <https://www.embopress.org/page/journal/14693178/authorguide#referencesformat>
- We note that Appendix Figure S7D and the panels of Appendix Figure S8 are currently not called out in the text.
- We note some irregularities in the background of p65 blot of Figure EV4G. Please provide a higher resolution image.
- We note that there are two lines intersecting the pages of Appendix Figures S4, S5, S6 and S7. Please rectify this.
- Please provide source data for Figure 6F and Appendix Fig S5A. Please see <https://www.embopress.org/page/journal/14693178/authorguide#sourcedata>
- Please upload Table 1 as 'Reagents and Tools Table' to the manuscript tracking system. Please see <https://www.embopress.org/page/journal/14693178/authorguide#textformat>
- We note that ORCID iD of Dr. Ping Zhang has not yet been linked. As of January 2016, new EMBO Press policy asks for all corresponding authors to link to their ORCID iDs. You can read about the change under "Authorship Guidelines" in the Guide to Authors here: <http://emboj.embopress.org/authorguide>

In order to link your ORCID iD to your account in our manuscript tracking system, please do the following:

1. Click the 'Modify Profile' link at the bottom of your homepage in our system.
2. On the next page you will see a box halfway down the page titled ORCID*. Below this box is red text reading 'To Register/Link to ORCID, click here'. Please follow that link: you will be taken to ORCID where you can log in to your account (or create an account if you don't have one)
3. You will then be asked to authorise Wiley to access your ORCID information. Once you have approved the linking, you will be brought back to our manuscript system.

We regret that we cannot do this linking on your behalf for security reasons.

- I have taken the liberty of performing some minor changes in the items below. Please take a look and confirm, or feel free to make further changes.

Title: CDC20 promotes bone formation via APC/C dependent ubiquitination and degradation of p65

Synopsis: This study reveals a cell-cycle independent function of CDC20 and identifies APC11CDC20 as a positive regulator of bone formation.

Abstract:

The E3 ubiquitin ligase complex CDC20-activated Anaphase-promoting complex/Cyclosome (APC/CCDC20) plays a critical role in governing mitotic progression by targeting key cell-cycle regulators for degradation. Cell division cycle protein 20 homolog (CDC20), the co-activator of APC/C, is required for full ubiquitin ligase activity. In addition to its well-known cell-cycle related functions, we demonstrate that CDC20 plays an essential role in osteogenic commitment of bone marrow mesenchymal stromal/stem cells (BMSCs). Cdc20 conditional knockout mice exhibit decreased bone formation and impaired bone regeneration after injury. Mechanistically, we discovered a functional interaction between the WD40 domain of CDC20 and the DNA-binding domain of p65. Moreover, CDC20 promotes the ubiquitination and degradation of p65 in an APC11-dependent manner. More importantly, knockdown of p65 rescues the bone loss in Cdc20 conditional knockout mice. Our current work reveals a cell-cycle independent function of CDC20, establish APC11CDC20 as a pivotal regulator for bone formation by governing the ubiquitination and degradation of p65, and may pave the way for treatment of bone-related diseases.

- Our production/data editors have asked you to clarify several points in the figure legends (see attached document). Please incorporate these changes in the attached word document and return it with track changes activated.

Thank you again for giving us to consider your manuscript for EMBO Reports, I look forward to your minor revision.

Kind regards,

Deniz Senyilmaz Tiebe

--

Deniz Senyilmaz Tiebe, PhD
Editor
EMBO Reports

Referee #1:

Yangge Du and collaborators have addressed thoroughly the concerns raised and suggestions made to improve the first version of their manuscript. This revised manuscript convincingly demonstrate that CDC20 is essential for bone formation and works at least in part through the control of proteasomal degradation of P65. This finding is both novel and important, and could have additional interesting implications in other tissues.

All concerns have been adequately addressed with the exception perhaps of the lack of characterization of the Sp7-iCre mouse line used to target osteoprogenitor cells in this study. The authors provided a link to the website of Biocytogen, the company that generated this line, but the website does not provide any characterization of the line nor any published article reporting such characterization. Du and colleagues also cited two publications in which authors have used the same Sp7-iCre line (from Biocytogen). Although these studies present abnormal bone phenotypes

(like in the current study of Du and colleagues), they do not demonstrate that the Sp7-iCre transgene targets specifically osteoprogenitors and does not have off-sites effects. It would be reassuring if the authors could provide data obtained by mating Sp7-iCre mice (generated by Biocytogene) with a Rosa26-LacZ or tdTomato reporter mouse lines to demonstrate that Sp7-iCre specifically targets osteoprogenitor cells in mice.

With the exception of this specific point, the work of Du and collaborators appears extremely well done, and the data robust.

Minor point: the age of the mice and type of bones used to generate bone sections presented in supplemental figures are not always mentioned.

Referee #2:

The authors have adequately addressed all my previous comments and further improved their manuscript.

Referee #3:

In the revised manuscript, the authors significantly strengthened their data in osteogenesis assays and Cdc20-dependent regulation of p65 stability. I would recommend it for publication.

Response to Editor and Reviewer

Ms. Ref. No.: EMBOR-2021-52576

Title: CDC20 promotes bone formation via APC/C dependent ubiquitination and degradation of p65

Dear Prof. Deniz Senyilmaz Tiebe,

We would like to express our sincere thanks to the editors and reviewers for the points concerning our manuscript. We found the suggestions of editors and reviewers to be very helpful and of great value for improving the quality of our manuscript. We have now made a revision of the paper. All the changes are highlighted in manuscript and a detailed point-to-point response were provided. We hope that our revised manuscript and supporting information will meet the high standard of *EMBO reports*.

- As per our format requirements, in the reference list, citations should be listed in alphabetical order and then chronologically, with the authors' surnames and initials inverted; where there are more than 10 authors on a paper, 10 will be listed, followed by 'et al.'. Please see <https://www.embopress.org/page/journal/14693178/authorguide#referencesformat>

Response: Thanks. We have rectified the format of references according to the requirements.

- We note that Appendix Figure S7D and the panels of Appendix Figure S8 are currently not called

out in the text.

Response: Thanks. We have added the “Appendix Figure S7D” in page 17 line 367 as follows.

“Then we conducted H&E staining and TUNEL staining of the liver, no discernable differences were found, suggesting that apoptosis was not induced in p65sh mice (Appendix Fig S7D).”

And we have changed “Appendix Figure S8” into “Appendix Figure S8A-Q” in page 18 line 394 to present the panels as follows.

“The statistical analyses of significantly different western blot lanes were presented in Appendix Fig S8A-Q.”

- We note some irregularities in the background of p65 blot of Figure EV4G. Please provide a higher resolution image.

Response: Thanks. We have provided a higher resolution image of p65 blot of Figure EV4G.

- We note that there are two lines intersecting the pages of Appendix Figures S4, S5, S6 and S7.

Please rectify this.

Response: Thanks. We have rectified this and uploaded Appendix Figures.

- Please provide source data for Figure 6F and Appendix Fig S5A. Please see <https://www.embopress.org/page/journal/14693178/authorguide#sourcedata>

Response: Thanks. We have provided source data of Figure 6F and Appendix Fig S5A, and uploaded this on the system.

Source Data

- Please upload Table 1 as 'Reagents and Tools Table' to the manuscript tracking system. Please see <https://www.embopress.org/page/journal/14693178/authorguide#textformat>

Response: Thanks. We have uploaded Table 1 as 'Reagents and Tools Table' to the manuscript

tracking system.

- We note that ORCID iD of Dr. Ping Zhang has not yet been linked. As of January 2016, new EMBO Press policy asks for all corresponding authors to link to their ORCID iDs. You can read about the change under "Authorship Guidelines" in the Guide to Authors here: <http://emboj.embopress.org/authorguide>

In order to link your ORCID iD to your account in our manuscript tracking system, please do the following:

1. Click the 'Modify Profile' link at the bottom of your homepage in our system.
2. On the next page you will see a box halfway down the page titled ORCID*. Below this box is red text reading 'To Register/Link to ORCID, click here'. Please follow that link: you will be taken to ORCID where you can log in to your account (or create an account if you don't have one)
3. You will then be asked to authorise Wiley to access your ORCID information. Once you have approved the linking, you will be brought back to our manuscript system.

We regret that we cannot do this linking on your behalf for security reasons.

Response: Thanks. We have revised it in the system.

• I have taken the liberty of performing some minor changes in the items below. Please take a look and confirm, or feel free to make further changes.

Title: CDC20 promotes bone formation via APC/C dependent ubiquitination and degradation of p65

Synopsis: This study reveals a cell-cycle independent function of CDC20 and identifies APC11CDC20 as a positive regulator of bone formation.

Abstract:

The E3 ubiquitin ligase complex CDC20-activated Anaphase-promoting complex/Cyclosome (APC/CCDC20) plays a critical role in governing mitotic progression by targeting key cell-cycle regulators for degradation. Cell division cycle protein 20 homolog (CDC20), the co-activator of APC/C, is required for full ubiquitin ligase activity. In addition to its well-known cell-cycle related functions, we demonstrate that CDC20 plays an essential role in osteogenic commitment of bone marrow mesenchymal stromal/stem cells (BMSCs). Cdc20 conditional knockout mice exhibit decreased bone formation and impaired bone regeneration after injury. Mechanistically, we discovered a functional interaction between the WD40 domain of CDC20 and the DNA-binding domain of p65. Moreover, CDC20 promotes the ubiquitination and degradation of p65 in an APC11-dependent manner. More importantly, knockdown of p65 rescues the bone loss in Cdc20 conditional knockout mice. Our current work reveals a cell-cycle independent function of CDC20, establish APC11CDC20 as a pivotal regulator for bone formation by governing the ubiquitination and degradation of p65, and may pave the way for treatment of bone-related diseases.

Response: Thanks. We confirmed the minor changes and revised them in the manuscript.

- Our production/data editors have asked you to clarify several points in the figure legends (see attached document). Please incorporate these changes in the attached word document and return it with track changes activated.

Response: Thanks. We have changed the points in the figure legends in page 52, line 1075 as follows.

“(K) The co-localization of CDC20 and p65 in hBMSCs. Scale bar: 20 μ m.”

Referee #1:

Yangge Du and collaborators have addressed thoroughly the concerns raised and suggestions made to improve the first version of their manuscript. This revised manuscript convincingly demonstrate that CDC20 is essential for bone formation and works at least in part through the control of proteasomal degradation of P65. This finding is both novel and important, and could have additional interesting implications in other tissues.

All concerns have been adequately addressed with the exception perhaps of the lack of characterization of the Sp7-iCre mouse line used to target osteoprogenitor cells in this study. The authors provided a link to the website of Biocytogen, the company that generated this line, but the website does not provide any characterization of the line nor any published article reporting such characterization. Du and colleagues also cited two publications in which authors have used the

same Sp7-iCre line (from Biocytogen). Although these studies present abnormal bone phenotypes (like in the current study of Du and colleagues), they do not demonstrate that the Sp7-iCre transgene targets specifically osteoprogenitors and does not have off-sites effects. It would be reassuring if the authors could provide data obtained by mating Sp7-iCre mice (generated by Biocytogene) with a Rosa26-LacZ or tdTomato reporter mouse lines to demonstrate that Sp7-iCre specifically targets osteoprogenitor cells in mice.

With the exception of this specific point, the work of Du and collaborators appears extremely well done, and the data robust.

Response: Thanks for your suggestions. For the development of the Sp7-iCre line from Biocytogen, an F2A-iCre sequence cassette was placed between the coding sequence of exon 2 and 3'UTR of the Sp7 gene in C57BL/6 ES cells according to the link of website we provided. In this strain, Cre recombinase expression is under the control of Sp7 promoter. When crossed with a strain containing a loxP site-flanked sequence of interest, Cre-mediated recombination results in deletion of the flanked sequence in Sp7 expressing cells.

In Supplemental Figure 5 of the article entitled “Increased PLEKHO1 within osteoblasts suppresses Smad-dependent BMP signaling to inhibit bone formation during aging” published on the *Aging Cell* Journal, the ROSA26-PCAG-STOP^{fllox}-*Smad1*-mCherry knock-in mice were intercrossed with the Sp7-iCre line from Biocytogen to generate *Osx/Smad1* mice in which *Smad1* was specifically overexpressed in osteoblasts. The representative fluorescence micrographs showing the co-localization of Smad1 (mCherry, red) + and ALP+ (green) cells at tibiae

cyosections from *Osx/Smad1* mice, suggesting that the knock-in exogenous *Smad1* gene was specifically expressed in osteoblasts. The *Smad1* mRNA levels in bone versus non-bone tissues from *Osx/Smad1* and *Osx-Cre* mice and mCherry+ cells (OBs) versus mCherry- cells (Non-OBs) from *Osx/Smad1* mice were provided, indicating the specific target of bone tissue to eliminate the off-sites effects.

The links of the article and Supplemental Figure were as follows. DOI: 10.1111/acle.12566.

<https://pubmed.ncbi.nlm.nih.gov/28083909/>

<https://onlinelibrary.wiley.com/action/downloadSupplement?doi=10.1111%2Facle.12566&file=acle12566-sup-0001-SupInfo.pdf>

In the *EMBO J* article “Ubiquitin-specific protease USP34 controls osteogenic differentiation and bone formation by regulating BMP2 signaling”, the *Sp7-Cre;Usp34^{f/f}* mice were generated from Biocytogen. In this article, the *Sp7-Cre* recombines efficiently in pre-osteoblasts and *Sp7-Cre;Usp34^{f/f}* mice exhibited normal mendelian inheritance and growth features. The results showed that the specific deletion of *Usp34* from pre-osteoblasts resulted in low bone mass and decreased osteoblast function in mice. The DOI of this article is 10.15252/emboj.201899398.

Minor point: the age of the mice and type of bones used to generate bone sections presented in supplemental figures are not always mentioned.

Response: Thanks. We have added the age of mice and type of bones of bone sections in supplemental figures.

Referee #2:

The authors have adequately addressed all my previous comments and further improved their manuscript.

Referee #3:

In the revised manuscript, the authors significantly strengthened their data in osteogenesis assays and Cdc20-dependent regulation of p65 stability. I would recommend it for publication.

Dear Dr. Zhou,

Thank you for submitting your revised manuscript. I have now looked at everything and all is fine. Therefore, I am very pleased to accept your manuscript for publication in EMBO Reports.

Congratulations on a nice work!

Kind regards,

Deniz Senyilmaz Tiebe

--

Deniz Senyilmaz Tiebe, PhD
Editor
EMBO Reports

--

At the end of this email I include important information about how to proceed. Please ensure that you take the time to read the information and complete and return the necessary forms to allow us to publish your manuscript as quickly as possible.

As part of the EMBO publication's Transparent Editorial Process, EMBO reports publishes online a Review Process File to accompany accepted manuscripts. As you are aware, this File will be published in conjunction with your paper and will include the referee reports, your point-by-point response and all pertinent correspondence relating to the manuscript.

If you do NOT want this File to be published, please inform the editorial office within 2 days, if you have not done so already, otherwise the File will be published by default [contact: emboreports@embo.org]. If you do opt out, the Review Process File link will point to the following statement: "No Review Process File is available with this article, as the authors have chosen not to make the review process public in this case."

Should you be planning a Press Release on your article, please get in contact with emboreports@wiley.com as early as possible, in order to coordinate publication and release dates.

Thank you again for your contribution to EMBO reports and congratulations on a successful publication. Please consider us again in the future for your most exciting work.

THINGS TO DO NOW:

You will receive proofs by e-mail approximately 2-3 weeks after all relevant files have been sent to our Production Office; you should return your corrections within 2 days of receiving the proofs.

Please inform us if there is likely to be any difficulty in reaching you at the above address at that time. Failure to meet our deadlines may result in a delay of publication, or publication without your

corrections.

All further communications concerning your paper should quote reference number EMBOR-2021-52576V3 and be addressed to emboreports@wiley.com.

Should you be planning a Press Release on your article, please get in contact with emboreports@wiley.com as early as possible, in order to coordinate publication and release dates.

Corresponding Author Name: Yongsheng Zhou, Ping Zhang

Journal Submitted to: EMBO reports

Manuscript Number: EMBOR-2021-52576